# Tree-Guided Diffusion Planner

**Hyeonseong Jeon**[1]  **Cheolhong Min**[1]  **Jaesik Park**[1,2]

[1]Department of Computer Science & Engineering, [2]Interdisciplinary Program of AI
Seoul National University

## Abstract

Planning with pretrained diffusion models has emerged as a promising approach for solving test-time guided control problems. Standard gradient guidance typically performs optimally under convex, differentiable reward landscapes. However, it shows substantially reduced effectiveness in real-world scenarios with non-convex objectives, non-differentiable constraints, and multi-reward structures. Furthermore, recent supervised planning approaches require task-specific training or value estimators, which limits test-time flexibility and zero-shot generalization. We propose a **T**ree-guided **D**iffusion **P**lanner (TDP), a zero-shot test-time planning framework that balances exploration and exploitation through structured trajectory generation. We frame test-time planning as a tree search problem using a bi-level sampling process: (1) diverse parent trajectories are produced via training-free particle guidance to encourage broad exploration, and (2) sub-trajectories are refined through fast conditional denoising guided by task objectives. TDP addresses the limitations of gradient guidance by exploring diverse trajectory regions and harnessing gradient information across this expanded solution space using only pretrained models and test-time reward signals. We evaluate TDP on three diverse tasks: maze gold-picking, robot arm block manipulation, and AntMaze multi-goal exploration. TDP consistently outperforms state-of-the-art approaches on all tasks. The project page can be found at: **tree-diffusion-planner.github.io**.

## 1  Introduction

Diffusion models offer a data-driven framework for planning, enabling the generation of coherent and expressive trajectories learning from offline demonstrations [1–4]. Compared to single-step model-free reinforcement learning (RL) methods [5, 6], diffusion planners are more effective for long-horizon planning by generating temporally extended trajectories through multi-step prediction. Without task-specific dynamics models, pretrained diffusion planners can be adapted for test-time planning through guidance functions that provide numerical scores for user requirements such as state conditions or physical constraints. Prior studies in diffusion guidance algorithm (*e.g.*, classifier guidance [7]) have been successfully incorporated to generate conditional trajectory samples given test-time reward signals [1, 8]. These guidance algorithms present an exploration-exploitation trade-off [9], balancing adherence to the pretrained model for feasibility against maximizing the external guide score, which may require exploring out-of-distribution trajectories.

As the main bottleneck in guided planning lies in the limited quality of trajectory samples produced by the pretrained models, prior works have primarily focused on improving general sample quality and mitigating planning artifacts, while the guidance algorithms themselves remain relatively underexplored. The majority of recent studies on diffusion planners emphasize advancing **supervised planning** capabilities [2–4, 10–12]. They are categorized into sequential [10, 11], hierarchical [3, 4], and fine-tuning [2, 12] approaches. These works improve the modeling of the underlying system dynamics from the static offline RL benchmarks. Recent works successfully solve challenging offline

39th Conference on Neural Information Processing Systems (NeurIPS 2025).

benchmarks by reformulating the training scheme, learning value estimator, and test-time scaling. Another recent research direction in diffusion planners aims to enhance **zero-shot planning** capabilities [13]. Test-time tasks are shifted from the training distribution, and planners are given access to a pretrained model along with dense reward signals to adapt to these unseen tasks effectively.

However, test-time planning capabilities are often evaluated in relatively simple optimization tasks, such as minimizing the distance to optimal trajectories or matching the outputs of pretrained classifiers trained on expert demonstrations [1, 4], and predominantly within in-distribution trajectory settings [2, 4]. Typical benchmarks include maze navigation tasks [10] where agents minimize distance to a single goal, or block stacking tasks [1] where a unique target configuration maximizes reward. These tasks generally involve convex optimization problems, where a unique global optimal trajectory maximizes a smooth guide function. In addition, similar to model-free RL methods [14], guided planning often relies on a pretrained value estimator trained on supervised trajectory data; however, collecting optimal trajectories for each new task is often impractical or infeasible, particularly for tasks with complex dynamics or limited simulation control.

In this work, we address a fundamental challenge of existing diffusion-based planners: while they excel at generating low-level action sequences, most real-world planning tasks require decision-making over high-level abstractions. These abstractions often introduce non-convex guide functions or non-differentiable constraints, making them incompatible with conventional test-time guidance methods that assume smooth, convex optimization landscapes. For example, maze navigation with intermediate goal bypassing [2] introduces a non-differentiable rule into the planning process. In multi-reward block stacking, the agent must reconcile multiple configuration-dependent reward signals to determine the most favorable block arrangement. These scenarios underscore the need for guided planning algorithms that flexibly accommodate complex test-time specifications while ensuring both trajectory feasibility and task-specific fitness. Recently, several training-free guidance algorithms have been proposed in the image domain [15–18], but their capabilities are typically restricted to smooth differentiable guide functions. Since test-time guide functions can take any form, there is a need for a flexible planning algorithm that can handle a broad class of guide functions.

We propose **T**ree-guided **D**iffusion **P**lanner (TDP), which formulates test-time planning as a tree search problem that balances exploration via diverse trajectory samples and exploitation via guided sub-trajectories. While pretrained diffusion planners model underlying system dynamics, TDP samples high-reward (*i.e.*, high guidance score for the test task) solution trajectories conditioned on the learned dynamics in a zero-shot manner. TDP equips the pretrained diffusion planner model with the ability to reason over higher-level objectives. TDP outperforms state-of-the-art planning methods on challenging tasks with non-convex guide functions and non-differentiable constraints across all test scenarios. TDP enables flexible task-aware planning without requiring expert demonstrations.

## 2 Related Work

**Existing Diffusion Planners.** Despite strong performance on standard offline benchmarks, existing diffusion planners face limitations in zero-shot planning:

- **Sequential approaches** [10, 11] explore one action at a time, which works well in single-goal convex tasks but struggles with multi-goal tasks where different goals have varying test-time priorities. In such settings, they often converge on local optima rather than discovering distant, high-priority goals. Furthermore, sequential approaches like MCTD [11] typically require training task-specific value estimators to guide their **single-step** decisions, limiting applicability to zero-shot scenarios where no task-specific training is available. In contrast, TDP performs **multi-step** exploration through diverse bi-level trajectory sampling, which better handles challenging multi-goal scenarios without requiring additional training components beyond the pretrained planner.
- **Hierarchical diffusion planners** [3, 4] rely on training-time supervision to learn sub-goal distributions. They perform well when both initial and goal states are given, but struggle on **unlabeled** zero-shot tasks such as the test-time gold-picking task [2]. In standard maze navigation benchmarks, Hierarchical Diffuser [4] tends to generate shortest-path trajectories when initial and goal states are specified. However, the gold-picking task poses a fundamentally different challenge: the goal is *hidden* and often misaligned with the shortest path.
- **Diffusion model predictive control** (D-MPC) [12] adapts to changing dynamics via **few-shot** fine-tuning with expert demonstrations, but struggles with unseen long-horizon tasks and complex

behaviors as standard dynamics models $p(s|a)$ struggle to capture long-context reward structures effectively. In contrast, TDP models the joint distribution $p(s, a)$ to enable solving long-horizon and multi-goal tasks and is a fully zero-shot planner that operates without test-time demonstrations.

**Training-free Guidance.** Training-free guidance methods leverage structural priors and domain knowledge for control without additional learning. Classical planners (*e.g.*, A*, potential fields) [19–23] compute feasible trajectories through graph search or geometric reasoning. In continuous domains, trajectory optimization and model predictive control [24, 25] refine actions iteratively using known dynamics. Local search techniques (*e.g.*, hill climbing), guided policy search, and reward shaping [26–28] serve as strong baselines for structured, domain-specific control problems but typically lack the flexibility to generalize beyond narrow task settings. In contrast, diffusion-based test-time planning targets complex tasks that need out-of-distribution generalization and adaptability.

**Tree-based Decision Making.** Tree structures naturally represent hierarchical sequential decisions. Trajectory Aggregation Tree (TAT) [29] mitigates artifacts in diffusion-generated trajectories by aggregating similar states near the initial state, but it struggles with complex long-horizon dependencies due to its limited aggregation depth early in the trajectory. Monte Carlo Tree Search [30–33] explores full-horizon trajectories via stochastic roll-outs, but its reliance on discrete actions and reward heuristics limits scalability in high-dimensional or continuous control tasks with external guidance. In contrast, TDP's bi-level tree framework integrates gradient-based guidance at both parent and child levels, enabling structured and adaptive planning under complex test-time objectives.

# 3 Background

## 3.1 Problem Setting

We consider the test-time reward maximization problem on a discrete-time dynamics system $s_{t+1} = f(s_t, a_t)$ via a pretrained planner model, where the agent has access to the user-defined guide function $\mathcal{J}(\tau)$, which indicates the fitness of the generated trajectory $\tau$. As per-timestep reward does not guarantee the optimality of a low-level action (*e.g.*, non-convex reward landscape), planning capability based on exploration is required to find the optimal trajectory $\hat{\tau}$ that maximizes $\mathcal{J}$. The agent must find an action sequence that maximizes the guide score within a limited number of steps:

$$\hat{\tau} = \hat{a}_{1:\hat{T}} = \underset{T, \, a_{1:T}}{\arg\max} \ \mathcal{J}(s_0, a_{1:T}) \quad \text{subject to} \quad T_{\text{pred}} \leq T \leq T_{\text{max}} \tag{1}$$

Planning horizon $T_{\text{pred}}$ is determined by the choice of planner model. Model-free RL methods with single step execution [5, 6] predict a single action at each timestep so $T_{\text{pred}} = 1$, whereas diffusion planner [1] predicts a sequence of actions $a_{1:T_{\text{pred}}}$ at once. As diffusion planners predict more future states, they benefit from capturing longer-term contextual information. For example, in a long-horizon multi-goal navigation task [34], an agent may encounter several intermediate suboptimal goals, but reaching a farther goal yields a substantially larger reward, requiring planning several steps rather than greedy pursuit of nearby rewards. Planning over longer horizons can enhance performance on more challenging tasks, particularly when future rewards are more significant [35, 36].

## 3.2 Test-time Guided Planning with Diffusion Models

The standard approach to guide diffusion planning in test time is to use naïve gradient guidance [18], which progressively refines the denoising process by combining the score estimate from the unconditional diffusion model with the auxiliary guide function [1, 4]. It approximates the reverse denoising process as Gaussian with small perturbation if the guidance distribution $h(\tau_i)$ is sufficiently smooth and the gradient of the guide function is time-independent:

$$\tilde{p}(\tau_{i-1}|\tau_i) \propto p_\theta(\tau_{i-1}|\tau_i)h(\tau_i) \approx \mathcal{N}(\tau_{i-1}; \mu^i + \alpha\Sigma^i g, \Sigma^i) \tag{2}$$

where $g = \nabla_\tau \log h(\tau_i)$ is the gradient of the guidance distribution [37], $\alpha$ is guidance strength, and $\mu, \Sigma$ are the mean and covariance of the pretrained reverse denoising process. On the other hand, classifier guidance (CG) [7] and classifier-free guidance (CFG) [38] are also broadly used in guided planning [3, 8, 39]. However, they require access to expert demonstration data to train either a time-dependent classifier model or a conditional diffusion planner model based on trajectory rewards

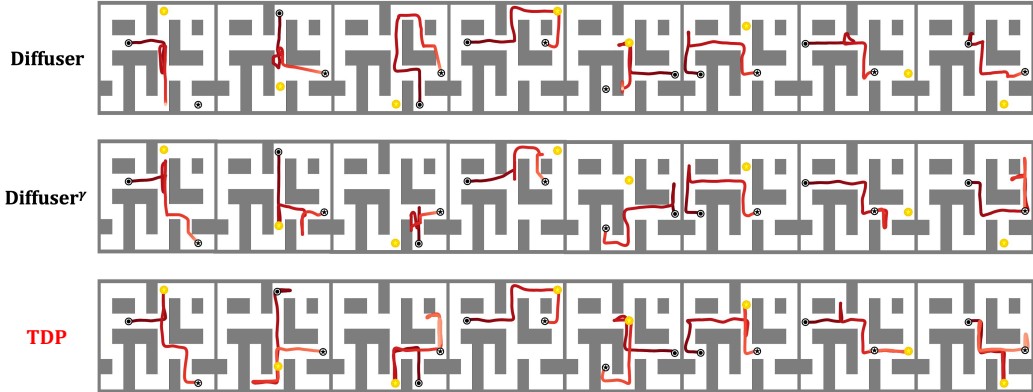

Figure 1: **Rollout Trajectories for the Gold-Picking in Maze2D.** In the Maze2D-Large environment [40], the agent must collect an additional gold objective (yellow) positioned off the shortest navigation path. Trajectories are generated by Diffuser [1] (with gradient guidance), Diffuser$^\gamma$ [29], and our method. See Sec. 5.2 for details.

for a given task. Despite their simple yet powerful architecture, it is expensive to extend CG and CFG in test-time as it requires collecting expert demonstrations for each new task and retraining the model. The ultimate goal of test-time guided planning is *adaptive* planning, identifying trajectories that satisfy user requirements using a pretrained diffusion planner without additional expert supervision.

### 3.3 Challenges in Guided Planning

Guided planning has primarily been evaluated on simple tasks where expert demonstrations are available or the guide function is convex and differentiable. As a result, prior work has focused on improving the sampling quality of the pretrained diffusion model, which was the main bottleneck for test-time guidance in these simplified settings [4, 41]. Despite the importance of improving trajectory sampling quality, there has been limited investigation into how guidance algorithms themselves adapt to increasing task complexity and respond to various forms of test-time objectives. In this section, we first study a fundamental challenge in guided planning: the exploitation-exploration trade-off, and then discuss the limitations of naïve gradient guidance, a standard test-time planning approach.

**Exploration-Exploitation Trade-off.** Test-time guided planning employs two distinct score functions: score estimate from the pretrained diffusion model and a user-defined guide score. Not only to generate a feasible trajectory but also to maximize its fitness, the agent is encouraged to balance the exploitation of the pretrained model and the exploration of novel trajectories. Gradient-based guidance typically requires selecting a guidance strength $\alpha$ to balance adherence to the guide signal and trajectory fidelity. However, $\alpha$ is highly task-dependent, and exhaustive tuning across tasks introduces significant overhead during evaluation. For example, Fig. 1 illustrates several gold-picking tasks within a fixed maze map. The optimal value of $\alpha$ varies across tasks, depending on how far the gold location deviates from the unconditional navigation trajectory, which favors the shortest path (see the supplement for more details).

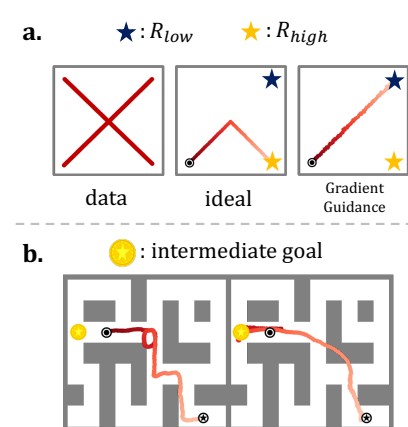

Figure 2: **Limitation of Gradient-based Guided Planning. a.** In-distribution preference. **b.** Naïve gradient guidance is incompatible with non-differentiable rules.

**In-distribution Trajectory Preference.** Diffusion models are capable of generating compositional behaviors [1, 42], but often get stuck in local optima due to insufficient exploration of the trajectory space. Most previous works on guided planning did not adequately address the exploration-exploitation trade-off, as they typically assume convex (or concave) guidance with a single optimal trajectory that globally maximizes the reward [2, 43]. However, real-world tasks often involve multiple objectives with different priorities, where the agent must explore the trajectory space to avoid sub-optimality. Pretrained diffusion planners tend to favor generating in-distribution trajectories that align with previously seen data, rather than discovering novel compositional solutions [4].

Consequently, gradient-based guidance algorithms do not effectively address this dilemma, as they often prioritize local optimal trajectories within the learned distribution (see Fig. 2-a).

**Non-differentiable Rule.** Gradient-based guidance algorithms face significant challenges in planning tasks with non-differentiable constraints. Since diffusion planners are trained on trajectories with fixed start and end states, they struggle to generate feasible paths that incorporate additional intermediate goals. For example, as described in Fig. 2 -b, navigation trajectories conditioned to test-time intermediate goal often result in suboptimal (left) or infeasible (right). This introduces a non-differentiable constraint from the planner's perspective, as the requirement to pass through a specific state imposes a discrete structural condition not reflected in the training distribution. Similar challenges arise in other domains of diffusion-based generation, such as enforcing chord progression in music generation [44] or satisfying chemical rules in molecule generation [45]. Both introduce hard constraints that are difficult to optimize with standard gradient-based methods.

# 4 Method

We introduce **T**ree-guided **D**iffusion **P**lanner (TDP), a zero-shot test-time planning framework leveraging a pretrained diffusion planner for adaptive trajectory generation. While naïve gradient guidance often converges to local optima due to limited gradient signals, TDP addresses this by combining diverse trajectory samples (*exploration*) with gradient-guided sub-trajectories (*exploitation*) to identify optimal solutions (see Fig. 3). This tree structure enables coverage over a broad range of reward landscapes. By branching into diverse regions, TDP increases the chance of finding globally high-reward solutions that naïve gradient methods may miss.

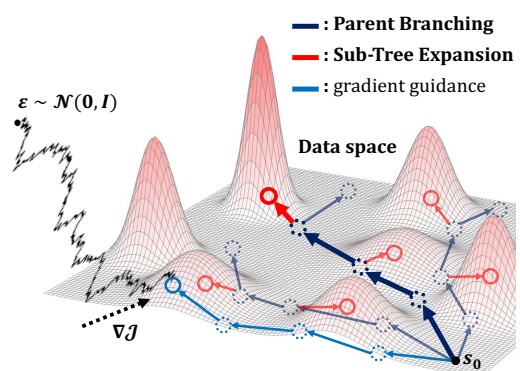

Figure 3: **Tree-guided Diffusion Planner (TDP).** TDP constructs a trajectory tree combining diverse parent trajectories (navy) and guided sub-trajectories (red) in the 2D data space. The 3D surface represents the reward landscape, with peaks indicating high-reward regions.

Appendix A, B outline the overall TDP pipeline and present the full algorithms. We detail the core modules of TDP: state decomposition (Sec. 4.1), parent branching (Sec. 4.2), and sub-tree expansion (Sec. 4.3).

## 4.1 State Decomposition

Given a guide function for the test-time task, states are autonomously decomposed based on gradient signals: *observation* states receive non-zero gradients, while *control* states receive none. This gradient-based criterion enables scalable and domain-agnostic decomposition at planning time. *Observation* states are directly steered by the guide function, whereas *control* states are unaffected by the guide function but govern the underlying system dynamics that support high-level objectives.

For instance, the KUKA robot arm environment provides state vectors containing multiple features such as robot joint angles and block positions. Since TDP operates as a zero-shot planner, it does not have prior knowledge of the state category for each feature of the state vector. Given a test-time block stacking task with a distance-based guide function, TDP autonomously categorizes each feature value in the state vector using Algorithm 1. It evaluates whether the gradient of the guide function with respect to the $i$th feature (*i.e.*, $\frac{\partial \mathcal{J}}{\partial s_i}$) is zero or non-zero. If non-zero, the $i$th feature is classified as an *observation* state; if zero, it is classified as a *control* state. Consequently, features related to robot physics are detected as *control* states. In contrast, block position (xy) features are detected as *observation* states, because the block-stacking guide function is only affected by the block positions.

## 4.2 Parent Branching

In the first phase of our bi-level planning framework, *control* states are applied to fixed-potential particle guidance (PG) [46] to explore diverse *control* trajectories. PG promotes diversity among generated samples within a batch. For instance, when moving a block to a target position, multiple

*control* trajectories can accomplish this task. Standard gradient-based approaches often suffer from in-distribution bias and limited exploration, constraining trajectory diversity. In contrast, TDP enhances exploration through this procedure, generating what we term parent trajectories.

Specifically, fixed-potential PG is implemented using the gradient of a radial basis function (RBF), denoted $\nabla\Phi$, which can be computed directly from all pairwise distances between *control* trajectories within a batch. Unlike conventional gradient guidance methods that pull samples toward high-reward regions, PG introduces repulsive forces that push samples apart in the data space. This leads to a broad coverage of dynamically feasible trajectories independent of task objectives. Although fixed-potential PG incurs some inference overhead from computing kernel values between all trajectory pairs, it is significantly more sample-efficient than learned-potential variants [46], while remaining training-free and modular. A single denoising step for parent branching is denoted as:

$$\boldsymbol{\mu}_{\text{control}}^{i-1} \leftarrow \boldsymbol{\mu}_{\text{control}}^{i} + \alpha_p \Sigma^i \nabla\Phi(\boldsymbol{\mu}_{\text{control}}^{i}), \quad \boldsymbol{\mu}_{\text{obs}}^{i-1} \leftarrow \boldsymbol{\mu}_{\text{obs}}^{i} + \alpha_g \Sigma^i \nabla\mathcal{J}(\boldsymbol{\mu}_{\text{obs}}^{i}), \tag{3}$$

where $\boldsymbol{\mu}_{\text{control}}^{i}$ and $\boldsymbol{\mu}_{\text{obs}}^{i}$ denote the *control* and *observation* components of the predicted mean of the denoising trajectory at timestep $i$. Particle guidance $\nabla\Phi(\boldsymbol{\mu}_{\text{control}})$ introduces repulsive updates among *control* states to diversify denoising paths. In contrast, gradient guidance $\nabla\mathcal{J}(\boldsymbol{\mu}_{\text{obs}})$ steers *observation* states toward task-relevant regions defined by the guide function. This enables a wide exploration in the *control* state space, which helps to discover diverse *observation* state configurations and exposes the planner to richer gradient signals from the guide function $\mathcal{J}$.

Notably, TDP formulates a single joint conditional distribution with an integrated guidance term. As shown in Eq. 2, the overall guidance distribution is defined as $h(\boldsymbol{\tau}_i) = h_{\text{gg}}(\boldsymbol{\tau}_i) \cdot h_{\text{pg}}(\boldsymbol{\tau}_i)$ where $h_{\text{gg}}(\cdot)$ denotes the gradient guidance component, and $h_{\text{pg}}(\cdot)$ denotes the particle guidance component. Since both components jointly condition the same pretrained reverse denoising process, the perturbed reverse denoising process is approximated as $\tilde{p}(\boldsymbol{\tau}_{i-1}|\boldsymbol{\tau}_i) \approx \mathcal{N}(\boldsymbol{\tau}_{i-1}; \mu^i + \alpha_{\text{TDP}}\Sigma^i g_{\text{TDP}}, \Sigma^i)$ where $g_{\text{TDP}} = \nabla_\tau \log(h_{\text{gg}}(\boldsymbol{\tau}_i) \cdot h_{\text{pg}}(\boldsymbol{\tau}_i)) = \nabla_\tau \log h_{\text{gg}}(\boldsymbol{\tau}_i) + \nabla_\tau \log h_{\text{pg}}(\boldsymbol{\tau}_i) = g_{\text{gg}} + g_{\text{pg}}$. Therefore,

$$\boldsymbol{\mu}^{i-1} \leftarrow \boldsymbol{\mu}^i + \alpha_{\text{TDP}}\Sigma^i g_{\text{TDP}} \tag{4}$$

denotes the integrated guidance term from Eq. 3, where $\boldsymbol{\mu}^{i-1} = [\boldsymbol{\mu}_{\text{control}}^{i-1}, \boldsymbol{\mu}_{\text{obs}}^{i-1}]$ and $g_{\text{TDP}} = g_{\text{gg}} + g_{\text{pg}} = \frac{1}{\alpha_{\text{TDP}}}(\alpha_p \nabla\Phi(\boldsymbol{\mu}_{\text{control}}^{i}) + \alpha_g \nabla\mathcal{J}(\boldsymbol{\mu}_{\text{obs}}^{i}))$. TDP employs a single integrated guidance term, which is the sum of the gradient guidance part ($g_{\text{gg}}$) and the particle guidance part ($g_{\text{pg}}$).

### 4.3 Sub-Tree Expansion

In the second phase, we apply fast denoising with reduced steps $N_f \ll N$, where $N$ is the original number of diffusion steps, to refine parent trajectories using task gradient signals. For each parent trajectory, we select a random branch site and generate a child trajectory by denoising from a partially noised version of the parent, conditioned on the preceding segment. Sub-tree expansion proceeds as:

$$\boldsymbol{\tau}_{\text{child}}^{N_f} \sim q_{N_f}(\boldsymbol{\tau}_{\text{parent}}, \boldsymbol{C}) \quad \text{where } \boldsymbol{C} = \{\boldsymbol{s}_k\}_{k=0}^{b} \text{ and } b \sim Unif(0, T_{\text{pred}}), \tag{5}$$

where $\boldsymbol{C}$ denotes the parent trajectory prefix, $q_{N_f}$ is the partial forward noising distribution with $N_f$ denoising steps, and $\boldsymbol{\tau}_{\text{child}}^{N_f}$ is the partially noised trajectory from which the child trajectory is denoised during sub-tree expansion. These child trajectories enable fine-grained local search around the parent branch, improving alignment with the guide signal. Sub-tree expansion offers two key advantages:

- Enhance **dynamic feasibility** of parent trajectories: Diverse parent trajectories benefit exploration, but perturbing the *control* states may lead to dynamically infeasible plans. During sub-tree expansion, perturbed *control* states are refined by a pretrained diffusion denoising process.

- Efficient **Local search**: Sub-Tree expansion refines *observation* states of parent trajectories with gradient guidance signal. Parent trajectories serve as initial points to guide child trajectories. Since parent trajectories are intended to cover a broad region of search space, local search conditioned on the parent trajectories is an efficient way to find better local optima.

**Why is bi-level sampling necessary?**  We investigate the role of our bi-level sampling framework in handling multi-reward structures, as illustrated in Fig. 3. We characterize problems with both local and global optima and demonstrate that bi-level trajectory generation avoids local optima. Consider

trajectory data in a learned subspace, where the pretrained diffusion planner maps Gaussian noise back to data space via the reverse denoising process. Given a test-time guide function defined as the sum of Gaussian components (as defined in Proposition 1), the following statements hold:

**Proposition 1.** *Initialization problem in gradient guidance with diffusion planner. Assume that the trajectory data $X \in \mathbb{R}^{H \times D}$ follows the Assumption 1 in [18], and given guide $\mathcal{J}(X) = \mathcal{J}_1(X) + \mathcal{J}_2(X)$ where $\mathcal{J}_1(X) = \exp\left(-\frac{1}{2\sigma_1^2}\|X - Av_1\|^2\right)$, $\mathcal{J}_2(X) = \exp\left(-\frac{1}{2\sigma_2^2}\|X - w_\perp\|^2\right)$, $w_\perp \perp \mathrm{span}(A)$, $v_1$ is the first eigenvector of $A^\top A$, $\sigma_1 < \sigma_2$, and $\mathbb{E}_{n \sim \mathcal{N}(0,I)}[\mathcal{J}_1(n)] \ll \mathbb{E}_{n \sim \mathcal{N}(0,I)}[\mathcal{J}_2(n)]$.*

    **a.** *If $X_0 \sim \mathcal{N}(0, I)$, then the guided sample $X_T$ converges to a local optimal solution.*

    **b.** *If $X_0 \sim q_{N_f}(\hat{X}^T, \Sigma^T)$ where $\hat{X}^T$ is an unconditional sample with small perturbation, then the guided sample $X_T$ converges to a global optimal solution.*

The proof is provided in Appendix C. Guided sampling initialized from standard Gaussian noise can be steered away from the subspace, converging to *off-subspace* local optima [18], as illustrated in Proposition 1-a. These samples may locally optimize the task objective but lie outside the learned data manifold, resulting in unrealistic or infeasible trajectories. This occurs because standard gradient guidance modifies the pretrained denoising process without preserving the underlying data structure, causing the sampling to drift toward regions that satisfy the guidance objective but violate the training distribution constraints. In practice, many test-time planning problems exhibit *off-subspace* optima—for example, in maze navigation, unseen obstacles can block the learned shortest path, making pretrained trajectory preferences suboptimal. In contrast, unconditional samples naturally remain close to the learned data subspace since they follow the original training distribution. When guided sampling is initialized from these *on-subspace* points, it can leverage both data structure and gradient information to converge to the global optimum, as demonstrated in Proposition 1-b.

TDP's bi-level planning framework enables flexible exploration through structural search over trajectories. During parent branching, gradient guidance can be optionally applied to steer the *observation* states, depending on task characteristics. Unconditional PG supports broad exploration of the data space, effective for discovering diverse solutions in under-specified or multi-goal settings. In contrast, conditional PG directs exploration toward regions aligned with the guide function, useful for convex objectives where a single high-quality solution is desired (*e.g.*, PNP tasks in Sec. 5.3). Gradient guidance in parent branching provides a prior for sub-tree expansion but may also limit the breadth of exploration in trajectory space. Each strategy offers complementary benefits: unconditional PG promotes diversity without task-specific priors, while conditional PG enables targeted search.

## 5 Experiments

We evaluate TDP across diverse zero-shot planning tasks featuring non-convex and non-differentiable objectives. Our experiments assess zero-shot planning capabilities and robustness when addressing unseen test-time objectives. All experimental hyperparameters are reported in Appendix D.

While existing diffusion planners (*e.g.*, MCTD [11], Hierarchical Diffuser [4]) excel on standard offline benchmarks, they suffer from zero-shot scenarios as discussed in Sec. 2. Consequently, we focus comparisons on recent zero-shot planning approaches, particularly Trajectory Aggregation Tree (TAT) [29], Monte-Carlo sampling [43], and stochastic sampling [13]. We design our benchmarks to challenge planners with unseen test-time objectives, in contrast to offline benchmarks that only test learned dynamics aligned with the training distribution. For completeness, we provide supplementary comparisons with sequential approaches (*i.e.*, MCTD [11], Diffusion-Forcing [10]) on standard maze benchmarks in Appendix J. Notably, despite being designed specifically for zero-shot planning, TDP still surpasses these sequential approaches on standard benchmarks.

### 5.1 Baselines and Ablations

- **Diffuser** [1]: Diffusion-based approach that plans by iteratively denoising complete trajectories.
- **AdaptDiffuser** [2]: Fine-tune pretrained Diffuser with synthetic expert demonstrations.
- **Diffuser$^\gamma$ (TAT)** [29]: Aggregate diffusion samples into a tree structure, bounding trajectory artifacts. Although TAT equips the pretrained Diffuser with better performance in offline RL tasks, its zero-shot planning capability is strictly constrained by standard diffusion sampling.

Table 1: **Result of Maze2d Gold-picking.** Mean performance of our method and baselines across 20 tasks (5 random seeds each). Four types of test maps are depicted in Appendix E. ± denotes standard error.

| Environment | | Diffuser | Diffuser$^\gamma$ | MCSS | MCSS+SS | TDP (w/o child) | TDP (w/o PG) | TDP |
|---|---|---|---|---|---|---|---|---|
| Maze2d | Medium | $10.1 \pm 2.5$ | $12.3 \pm 2.6$ | $17.2 \pm 3.4$ | $17.4 \pm 3.2$ | $19.0 \pm 3.5$ | $39.1 \pm 4.2$ | $\mathbf{39.8} \pm 4.2$ |
| Maze2d | Large | $4.3 \pm 1.8$ | $9.3 \pm 2.6$ | $25.0 \pm 3.8$ | $21.2 \pm 3.5$ | $30.4 \pm 4.1$ | $41.1 \pm 4.2$ | $\mathbf{47.6} \pm 4.1$ |
| **Single-task Average** | | 6.8 | 10.8 | 21.1 | 19.3 | 24.7 | 40.1 | **43.7** |
| Multi2d | Medium | $7.7 \pm 2.4$ | $8.6 \pm 2.4$ | $32.3 \pm 3.9$ | $29.2 \pm 3.6$ | $35.3 \pm 4.3$ | $\mathbf{75.9} \pm 3.0$ | $74.7 \pm 3.0$ |
| Multi2d | Large | $9.9 \pm 2.6$ | $23.1 \pm 3.6$ | $57.5 \pm 4.0$ | $58.0 \pm 3.8$ | $59.1 \pm 3.9$ | $64.9 \pm 3.7$ | $\mathbf{70.0} \pm 3.5$ |
| **Multi-task Average** | | 8.8 | 15.9 | 44.9 | 43.6 | 47.2 | 70.4 | **72.4** |

- **Monte-Carlo Sampling with Selection (MCSS)** [43]: Sample multiple trajectories from the pretrained Diffuser and select the best trajectory based on the guidance score. Monte Carlo sampling methods effectively explore the solution space and approximate optimal trajectories [47].

- **Stochastic Sampling (MCSS+SS)** [13]: MCMC-based, training-free guided planning method requiring $M$ times more computation than MCSS, where $M$ is the number of iterations in the inner loop of diffusion sampling. Following [13], we set $M = 4$ in all our experiments.

- **TDP (w/o child)**: Remove the Sub-Tree Expansion phase from TDP, relying solely on parent trajectories generated by conditional PG sampling without further refinement through sub-trajectory sampling. This isolates the contribution of parent trajectories to overall planning performance.

- **TDP (w/o PG)**: Ablate the particle guidance step in Parent Branching, relying solely on gradient guidance to generate parent trajectories. This highlights PG's role in producing diverse parents that serve as initialization points for Sub-Tree Expansion.

## 5.2 Maze2d Gold-picking

We extend the single gold-picking example [2] in the Maze2D environment [40] to a multi-task benchmark. The agent is initialized at a random position in the maze and has to find the gold at least once before it reaches the final goal position. As discussed in Sec. 3.3, the gold-picking task is a planning problem with a test-time non-differentiable constraint, where the agent must generate a feasible trajectory that satisfies an initial state, a final goal state, and an intermediate target (the gold position). In addition, the task is a *black-box* problem that requires inferring the intermediate goal (*i.e.*, gold) location using only a distance-based guide function, without access to the gold's exact position. To address this setting with diffusion planners, prior work [2] proposes an approximate, distance-based guide function $\mathcal{J}(\boldsymbol{\tau}) = -\sum_{i=1}^{T_{\text{pred}}} \|\boldsymbol{s}_i - \boldsymbol{s}_{\text{gold}}\|$, where $\boldsymbol{s}_{\text{gold}}$ denotes the gold position. While this approximate function is used for the gradient guidance sampling, the true guide function $\mathcal{J}(\boldsymbol{\tau}) = -\min_{i \in \{1, \cdots, T_{\text{pred}}\}} \|\boldsymbol{s}_i - \boldsymbol{s}_{\text{gold}}\|$ is used for selecting the best one from the generated candidates. We report the performance of TDP and baselines in Table 1. TDP consistently outperforms both Diffuser$^\gamma$ and MCSS across single- and multi-task settings. Notably, even TDP (w/o child) achieves approx. a 7% performance improvement over MCSS overall. TDP generates farther sub-trajectories through sub-tree expansion, allowing the planner to localize the gold position better and collect stronger gradient signals from the surrounding region to guide the trajectory effectively. This bi-level trajectory sampling approach enables the discovery of the *hidden* gold location within the map.

## 5.3 KUKA Robot Arm Manipulation

We evaluate the test-time planning performance of TDP and baselines on robotic arm manipulation tasks. Diffusion planners are pretrained on arbitrary block stacking demonstrations collected from PDDLStream [48] and are typically evaluated on downstream tasks such as conditional stacking [1] and pick-and-place [2], where test-time goals are specified to the planner. Both tasks aim to place randomly initialized blocks into their corresponding target locations in a predetermined order. The pick-and-place task is more challenging because it requires placing each block at a unique target without access to expert demonstrations. To better isolate test-time planning capability, we evaluate a variant of the conditional stacking task [2] without using the pretrained classifier on expert data. We refer to this variant as PNP (*stack*), and the original task is denoted as PNP (*place*). For both tasks, the pretrained diffusion planner is guided to generate 4 sequential manipulation trajectories,

one for each block, toward its target location $s_{\text{target}}$, using a naïve guidance objective defined as $\mathcal{J}(\boldsymbol{\tau}) = -\sum_i \|s_i - s_{\text{target}}\|$. As shown in Table 2, TDP achieves an average improvement of 10% over MCSS and 20% over TAT in two PNP tasks, demonstrating strong generalization to diverse task configurations. Notably, TDP (w/o PG) outperforms MCSS by 18% on PNP (*place*), underscoring the role of our sub-trajectory refinement mechanism in such out-of-distribution planning scenarios.

Moreover, we carefully design a more challenging test-time manipulation task which extends PNP (*place*), namely pick-and-*where-to*-place (PNWP). As shown in Fig. 4, the agent must find a global optimal trajectory given non-concave guide function $\mathcal{J}(\boldsymbol{\tau}) = -\sum_i c_1 \|s_i - s_{\text{local}}\| - c_2 \|s_i - s_{\text{mid}}\| + c_3 \|s_i - s_{\text{global}}\|$ where $c_1, c_2, c_3$ are positive constants and $s_{\text{local}}, s_{\text{mid}}, s_{\text{global}}$ are the local, middle, and global target positions, respectively. Both $s_{\text{global}}$ and $s_{\text{local}}$ are equidistant from the robot arm's attachment point. Since the local optimum has a wide peak while the global optimum has a narrow peak, agents easily get trapped in local optima without sufficient exploration. While PNP requires fitting blocks into a target configuration, PNWP challenges planners to distinguish between globally optimal and suboptimal arrangements.

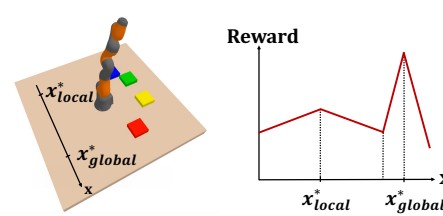

Figure 4: **Pick-and-*Where-to*-Place (PNWP).** PNWP evaluates the agent's exploration capacity in the robot arm manipulation environment. The agent must infer suitable placement locations for each block based on the reward distribution and plan corresponding pick-and-place actions.

Table 2: **Result of Robot Arm Manipulation.** Mean performance of baselines, ablations, and our method. Diffuser uses 1 sample, while others use samples $\in \{6, 12, 18, 24\}$ with 100 seeds. $\pm$ denotes standard error.

| Environment | Diffuser | Diffuser$^\gamma$ | AdaptDiffuser | MCSS | MCSS+SS | TDP (w/o child) | TDP (w/o PG) | TDP |
|---|---|---|---|---|---|---|---|---|
| PNWP | $31.13 \pm 0.07$ | $34.72 \pm 0.09$ | $39.72 \pm 0.08$ | $35.69 \pm 0.07$ | $36.24 \pm 0.09$ | $35.53 \pm 0.08$ | $66.63 \pm 0.15$ | $\mathbf{66.81} \pm 0.17$ |
| PNP (*stack*) | $51.5 \pm 0.08$ | $60.08 \pm 0.16$ | $60.54 \pm 0.18$ | $59.91 \pm 0.19$ | $56.8 \pm 0.16$ | $60.00 \pm 0.19$ | $59.42 \pm 0.19$ | $\mathbf{61.17} \pm 0.24$ |
| PNP (*place*) | $21.31 \pm 0.05$ | $21.44 \pm 0.10$ | $36.17 \pm 0.11$ | $31.37 \pm 0.10$ | $35.5 \pm 0.19$ | $32.19 \pm 0.10$ | $\mathbf{36.94} \pm 0.13$ | $\mathbf{36.94} \pm 0.13$ |
| **PNP Average** | 36.41 | 40.76 | 48.36 | 45.64 | 46.15 | 46.10 | 48.18 | **49.06** |

We report the performance of TDP, baselines, and ablations in Table 2. Mono-level guided sampling methods (*i.e.*, AdaptDiffuser, MCSS(+SS), TAT, and TDP (w/o child)) tend to converge to local optima, often stacking all blocks at a single position, since a landscape of local optima is spread out in a broader range as shown in Fig. 4. In contrast, bi-level sampling approaches (*i.e.*, TDP and TDP (w/o PG)), which combine parent branching and sub-trajectory refinement, are better able to identify globally optimal placements consistently. Notably, TDP outperforms AdaptDiffuser both on the standard benchmark (PNP) and on the custom task (PNWP). This demonstrates that TDP's test-time scalability and generalization enable zero-shot planning that surpasses the training-per-task approach.

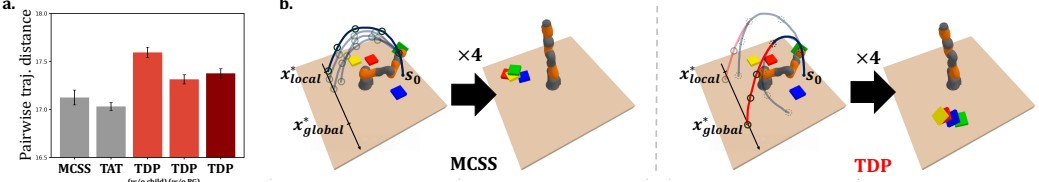

Figure 5: **Diverse Trajectory Generation. a.** Mean pairwise distance computed over 32 trajectories, averaged across 100 planning seeds in PNWP. Error bars indicate standard error. **b.** Visualization of trajectory generation process and rollout results of MCSS and TDP.

The key advantage of our method over the baselines comes from employing unconditional PG in the parent branching phase, enabling broad exploration of the trajectory space. Both TDP and its ablations produce diverse trajectories (see Fig. 5-a); however, TDP (w/o child) often fails to find the global optimal placement due to the use of conditional PG in parent branching, which can bias samples toward local optima. The mean pairwise trajectory distance decreases when PG is combined with sub-tree expansion, as the generated sub-trajectories share segments with their corresponding parent trajectories. Unconditional PG enables the generation of diverse, gradient-free parent trajectories. When combined with sub-tree expansion guided by the objective, this bi-level strategy supports compositional solutions that successfully stack blocks into global optimal placement (see Fig. 5-b). More detailed information on the evaluation metrics among these tasks is available in Appendix F.

## 5.4 AntMaze Multi-goal Exploration

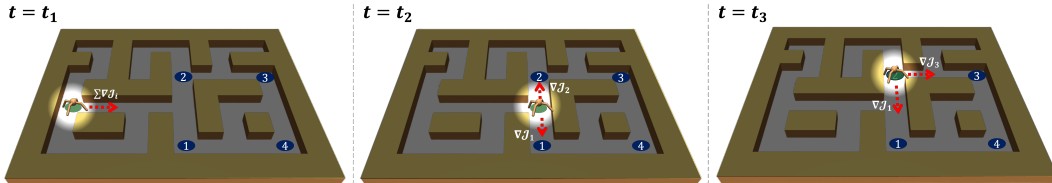

Figure 6: **AntMaze Multi-goal Exploration.** A priority-aware multi-goal exploration designed to evaluate exploration in AntMaze locomotion planning. A diffusion planner predicts the next $T_{\text{pred}} = 64$ steps (bright areas on the map) using a net guide signal from multiple goals, with a maximum horizon of 2000 steps.

We finally evaluate test-time multi-goal exploration capability on AntMaze [40], which is more challenging than Maze2D due to its high-dimensional observation space for controlling the embodied agent. Its complexity causes the pretrained diffusion planner to predict shorter horizons than the full trajectory length [49]. We design a new multi-goal task in which the agent must visit all goal positions in the correct order of priority, as specified by the guide function $\mathcal{J}(\boldsymbol{\tau}) = \sum_{g \in \mathcal{G}} h_g \cdot \exp\left(-\sum_{i \le T_{\text{pred}}} \|\boldsymbol{s}_i - \boldsymbol{s}_g\|^2 / \sigma_g^2\right)$ where $\mathcal{G}$ is the set of goal positions and $h_g$, $\sigma_g$ are parameters for gaussian guide function at each goal $g \in \mathcal{G}$. For example, the agent in Fig. 6 first visits goal $g_2$ at $t = t_3$. If it subsequently visits $g_1$, $g_4$, and $g_3$ after $t = t_3$, it successfully reaches all four goals in the sequence $g_2 \rightarrow g_1 \rightarrow g_4 \rightarrow g_3$. However, two precedence rules are violated ($g_2 \rightarrow g_1$, $g_4 \rightarrow g_3$) while the remaining four ($g_2 \rightarrow g_4$, $g_2 \rightarrow g_3$, $g_1 \rightarrow g_4$, and $g_1 \rightarrow g_3$) are satisfied. In this scenario, the agent achieves a goal completion score of 4/4 but only 4/6 priority sequence match accuracy. Maximum accuracy of 6/6 can only be achieved by visiting all goals in the correct prioritized order.

Table 3: **Result of AntMaze Multi-goal Exploration.** Mean performance of baselines, ablations, and our method. Diffuser uses 1 sample, while others use samples $\in \{32, 64, 128, 256\}$ with 100 seeds. We report three metrics: number of found goals, sequence match score, and average timesteps per goal. $\pm$ denotes standard error.

| Metric | Diffuser | Diffuser$^\gamma$ | MCSS | MCSS+SS | TDP (w/o child) | TDP (w/o PG) | TDP |
|---|---|---|---|---|---|---|---|
| # found goals ↑ | $1.5 \pm 0.3$ | $12.4 \pm 0.8$ | $61.3 \pm 2.0$ | $62.8 \pm 1.9$ | $65.8 \pm 1.9$ | $64.6 \pm 1.9$ | $\mathbf{66.1} \pm 1.9$ |
| sequence match ↑ | $1 \pm 0.56$ | $1.2 \pm 0.27$ | $30.2 \pm 1.3$ | $31.2 \pm 1.3$ | $\mathbf{33.8} \pm 1.5$ | $32.7 \pm 1.4$ | $\mathbf{33.8} \pm 1.4$ |
| # timesteps per goal ↓ | 5333.3 | 4020.1 | 612.1 | 604.4 | 574.3 | 578.3 | $\mathbf{558.4}$ |

We report the performance of TDP, baselines, and ablations in Table 3 using three metrics, with detailed definitions in Appendix F. TDP achieves about 11% improvements over MCSS in both the number of found goals and the sequence match score, while also reducing timesteps per goal. The ablations highlight complementary roles of the components: when PG is removed (TDP (w/o PG)), all three metrics degrade, confirming that PG is essential for both goal discovery and sequence alignment. In contrast, removing child branching (TDP (w/o child)) maintains performance on the first two metrics but requires more timesteps per goal, suggesting less efficient exploration in the environment.

# 6 Conclusion

In summary, we propose TDP, a flexible test-time planning framework that leverages a pretrained diffusion planner via a bi-level trajectory-sampling process without training. By balancing trajectory diversity and gradient-guided refinement via a branching structure of sampled trajectories, our method addresses key limitations of conventional test-time-guided planning. Empirical results across both structured and compositional manipulation tasks demonstrate consistent performance gains over existing baselines, particularly in scenarios that demand out-of-distribution generalization. Our experiments also highlight the robustness of the framework in handling non-convex, multi-objective guidance and non-differentiable constraint problems, where naïve gradient guided methods often fail.

**Limitation and Future Work.** While TDP outperforms existing planning approaches across a suite of challenging test-time control tasks, our bi-level trajectory generation process incurs additional computational cost due to the expanded search in trajectory space and the pairwise trajectory distance calculations. We analyze the additional computational time required for the two PNP tasks and the PNWP task in Appendix G. Future work may explore more efficient search strategies or learned priors to reduce overhead while ensuring sufficient exploration and preserving planning performance.

## Acknowledgements

This work was supported by IITP grant (RS-2021-II211343: AI Graduate School Program at Seoul National Univ. (5%) and RS-2025-25442338: AI Star Fellowship Support Program (Seoul National Univ.) (60%)) and NRF grant (No.2023R1A1C200781211 (35%)) funded by the Korea government (MSIT).

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

# A  Tree-guided Diffusion Planner

**Step 1. Parent Branching**   In the first phase of bi-level sampling, $N$ parent trajectories are generated via Algorithm 2. $N$ denotes the number of trajectories sampled in a batch.

**Step 2. Sub-Tree Expansion**   In the second phase of bi-level sampling, $N$ child trajectories are generated via Algorithm 3. Each child trajectory is generated from a parent trajectory, and the branching site is randomly chosen among intermediate states in the parent trajectory.

**Step 3.  Leaf Evaluation**   Now we construct a trajectory tree consisting of parent and child trajectories. The root node of the tree corresponds to the initial state, and the tree has $2N$ leaf nodes. Each leaf node represents a complete trajectory and is associated with a guide score of the path from the root. The path to the leaf with the highest score is selected as the final solution trajectory.

**Step 4. Action Execution**   TDP supports both open-loop and closed-loop planning. In open-loop planning, the agent executes the entire solution trajectory as planned. In closed-loop planning, the agent executes only the first action of the planned trajectory, then replans by repeating Step 1-3 at each timestep.

# B  Algorithms

---

**Algorithm 1 State Decomposition (SD)**

---

1: **Input:** $\mathcal{J}$, trajectory $\boldsymbol{\tau}$
2: $W :=$ number of features in state vector of $\boldsymbol{\tau}$
3: $(\boldsymbol{s}_1, \boldsymbol{s}_2, \dots \boldsymbol{s}_W) := \boldsymbol{\tau}$
4: $l_{\text{control}} \leftarrow [], l_{\text{obs}} \leftarrow []$
5: **for** $i = 1$ to $W$ **do**
6:     **if** $\frac{\partial \mathcal{J}}{\partial \boldsymbol{s}_i} == 0$ **then**
7:         Append $\boldsymbol{s}_i$ to $l_{\text{control}}$
8:     **else**
9:         Append $\boldsymbol{s}_i$ to $l_{\text{obs}}$
10:     **end if**
11: **end for**
12: **return** $[l_{\text{control}}, l_{\text{obs}}]$

---

**Algorithm 2 Parent Branching**

---

1: **Input:** $\mu_\theta$, $\Sigma^i$, $N$, $\mathcal{J}$, scales $(\alpha_p, \alpha_g)$, particle guidance kernel $\Phi(\cdot)$, condition $C$
2: Initialize plan $\boldsymbol{\tau}_{\text{parent}}^N \sim \mathcal{N}(0, I)$
3: **for** $i = N$ to 1 **do**
4:     $\mu^i \leftarrow \mu_\theta(\boldsymbol{\tau}_{\text{parent}}^i)$
5:     // state decomposition
6:     $[\mu_{\text{control}}^i, \mu_{\text{obs}}^i] \leftarrow \mathbf{SD}(\mathcal{J}, \mu^i)$
7:     // particle guidance
8:     $\mu_{\text{control}}^{i-1} \leftarrow \mu_{\text{control}}^i + \alpha_p \Sigma^i \nabla \Phi(\mu_{\text{control}}^i)$
9:     // gradient guidance
10:     $\mu_{\text{obs}}^{i-1} \leftarrow \mu_{\text{obs}}^i + \alpha_g \Sigma^i \nabla \mathcal{J}(\mu_{\text{obs}}^i)$
11:     $\mu_{\text{parent}}^{i-1} \leftarrow [\mu_{\text{control}}^{i-1}, \mu_{\text{obs}}^{i-1}]$
12:     $\boldsymbol{\tau}_{\text{parent}}^{i-1} \sim \mathcal{N}(\mu_{\text{parent}}^{i-1}, \Sigma^i)$
13:     Constrain $\boldsymbol{\tau}_{\text{parent}}^{i-1}$ with $C$
14: **end for**
15: **return** $\boldsymbol{\tau}_{\text{parent}}^0$

---

**Algorithm 3 Sub-Tree Expansion**

---

1: **Input:** $\mu_\theta$, $\Sigma^i$, $N$, $\mathcal{J}$, $\alpha_g$, $C$, parent trajectories $\boldsymbol{\tau}_{\text{parent}}$, number of samples $B$, planning horizon $T$, fast diffusion steps $N_f \ll N$
2: **for** $j = 1$ to $B$ **do**
3:     // random branch site
4:     $b_j \sim Uniform([0, T))$
5:     Extract first $b_j$ states from $\boldsymbol{\tau}_{\text{parent},j}$
6:     $C \leftarrow \{s_k\}_{k=0}^{b_j}$
7:     // perturb parent traj.
8:     $\boldsymbol{\tau}_{\text{child},j}^{N_f} \leftarrow q_{N_f}(\boldsymbol{\tau}_{\text{parent},j}, C)$
9:     // fast planning
10:     **for** $i = N_f$ to 1 **do**
11:         $\mu^i \leftarrow \mu_\theta(\boldsymbol{\tau}_{\text{child},j}^i)$
12:         $\boldsymbol{\tau}_{\text{child},j}^{i-1} \sim \mathcal{N}(\mu^i + \alpha_g \Sigma^i \nabla \mathcal{J}(\mu^i), \Sigma^i)$
13:         Constrain $\boldsymbol{\tau}_{\text{child},j}^{i-1}$ with $C$
14:     **end for**
15: **end for**
16: **return** $\boldsymbol{\tau}_{\text{child}}^0$

---

## C   Proof of Proposition 1

Given guide $\mathcal{J}(X) = \mathcal{J}_1(X) + \mathcal{J}_2(X)$, orthogonal gradient $\nabla_\perp \mathcal{J}(X)$ to the subspace spanned by $A$ is calculated as follows:

$$\nabla_\perp \mathcal{J}(X) = -\left[\frac{1}{\sigma_1^2}(I - AA^\top)(X - Av_1)\mathcal{J}_1(X) + \frac{1}{\sigma_2^2}(X - w_\perp)\mathcal{J}_2(X)\right] \tag{6}$$

We examine the orthogonal reverse process applying the orthogonal gradient guidance:

$$dX_{t,\perp} = \left(\frac{1}{2} - \frac{1}{h(T-t)}\right)X_{t,\perp}dt + \nabla_\perp f(X_t)dt + (I - AA^\top)d\overline{W}_t \tag{7}$$

where the conditional distribution $X_t|X_0$ in forward process is Gaussian, *i.e.*, $\mathcal{N}(\alpha(t)x_0, h(t)I)$ where $h(t) = 1 - \alpha^2(t) = 1 - \exp(-\sqrt{t})$ [18]. First we consider the case when $X_0 \sim \mathcal{N}(0, I)$. The orthogonal gradient can be approximated as $\nabla \mathcal{J}_\perp(X) \approx -\frac{1}{\sigma_2^2}(X_\perp - w_\perp)\mathcal{J}_2(X)$ by assumption. As $\mathcal{J}_2(X) \geq \mathbb{E}_{X_0}[\mathcal{J}_2(X_0)] =: b_0 > 0$ and taking expectation, it is derived by linear ODE $d\mathbb{E}[X_{t,\perp}] = \left[\left(\gamma(t) - \frac{b_0}{\sigma_2^2}\right)\mathbb{E}[X_{t,\perp}] + \frac{b_0}{\sigma_2^2}w_\perp\right]dt$. Solving this ODE, we get lower bound of the expectation of displacement of the final state $\mathbb{E}[|X_{T,\perp}|] \geq \exp\left(-\Phi(T,0)\right)\left(\frac{b_0}{\sigma_2^2}\int_0^T \exp\left(\Phi(s,0)\right)ds\right)|w_\perp|$ where $\Phi(t,s) = \int_s^t \left(\gamma(u) - \frac{b_0}{\sigma_2^2}\right)du$. Since the coefficient of direction vector $w_\perp$ is a tractable positive constant, the final state is always *off-subspace*. The final state converges linearly as the given guide function satisfies the PL condition and the Lipschitz property [50]. However, the global optimal solution lies in the subspace spanned by $A$, guided sample converges to local optimal solution which is orthogonal to the subspace.

Second, we consider the case when $X_0 \sim q_{N_f}(\hat{X}^T, \Sigma^T)$. As the small perturbation is added to the unconditional sample in the subspace, the orthogonal gradient can be approximated as $\nabla \mathcal{J}_\perp(X) \approx -\frac{1}{\sigma_1^2}(I - AA^\top)(X - Av_1)\mathcal{J}_1(X) = -\frac{1}{\sigma_1^2}X_\perp \mathcal{J}_1(X)$. From Eq. 7, we get linear ODE $d\mathbb{E}[X_{t,\perp}] = \left[\left(\gamma(t) - \frac{1}{\sigma_1^2}\mathcal{J}_1(X)\right)\mathbb{E}[X_{t,\perp}]\right]dt$. As $\mathbb{E}[X_{0,\perp}] = 0$, we have $\mathbb{E}[X_{T,\perp}] = 0$, which means the final state is *on-subspace*. Similar to the previous case, the final state converges linearly to the global optimal solution lying on the subspace as the given guide function satisfies the PL condition and Lipschitz property [50].

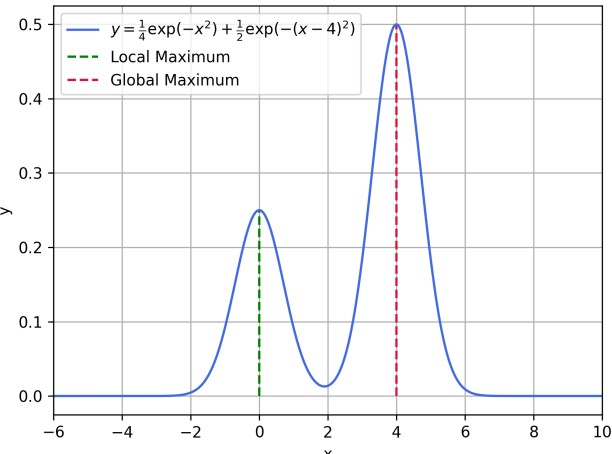

Figure 7: **1D Example of Local&Global optimum existing reward problem.** When gradient guidance is initialized at $x = -2$, it converges to a local maximum at $x = 0$. This example highlights the importance of using multiple initial points when applying gradient-based guidance, which can improve the probability of reaching the global optimum under a test-time guide function.

# D   Hyperparameters

In Maze2D gold-picking tasks, we use the same number of diffusion steps for both parent trajectory generation and sub-tree expansion, as any state within a valid maze cell can serve as a feasible trajectory starting point. For fair comparison, we evaluate MCSS and TAT with twice the number of samples (256) to account for their single-stage sampling design. All experiments were conducted using a single NVIDIA GeForce RTX 3090 GPU.

Table 4: **Hyperparameters of three tasks.**

| Task | Name | Value |
|---|---|---|
| **Maze2D Gold-picking** | maze2d-medium planning horizon $T_{\text{pred}}$ | 256 |
| | maze2d-medium maximum steps $T_{\text{max}}$ | 600 |
| | maze2d-large planning horizon $T_{\text{pred}}$ | 384 |
| | maze2d-large maximum steps $T_{\text{max}}$ | 800 |
| | Threshold distance | 0.3 |
| | gradient guidance strength $\alpha_g$ | 62.5 |
| | particle guidance strength $\alpha_p$ | 0.1 |
| | diffusion steps $N = N_f$ | 256 |
| | Number of samples $B$ | 128 |
| **Kuka Robot Arm Manipulation** | planning horizon $T_{\text{pred}}$ | 256 |
| | maximum steps $T_{\text{max}}$ | 800 |
| | guide function parameters $c_1, c_2, c_3$ | 1.0, 1.5, 2.0 |
| | peak width ratio ($\|s_{\text{mid}} - s_{\text{local}}\| : \|s_{\text{mid}} - s_{\text{global}}\|$) | 3:1 |
| | Threshold distance (PNP (*stack*), PNP (*place*)) | 0.2 |
| | Threshold distance (PNWP) | 0.4 |
| | gradient guidance strength $\alpha_g$ | 100 |
| | particle guidance strength $\alpha_p$ (PNP (*stack*)) | 0.25 |
| | particle guidance strength $\alpha_p$ (PNP (*place*)) | 0.50 |
| | diffusion steps $N$ | 1000 |
| | fast diffusion steps $N_f$ | 100 |
| **AntMaze Multi-goal Exploration** | planning horizon $T_{\text{pred}}$ | 64 |
| | maximum steps $T_{\text{max}}$ | 2000 |
| | Threshold distance | 0.1 |
| | gradient guidance strength $\alpha_g$ | 0.1 |
| | particle guidance strength $\alpha_p$ | 0.1 |
| | first goal configuration $(h_{g_1}, \sigma_{g_1})$ | (4.0, 0.05) |
| | second goal configuration $(h_{g_2}, \sigma_{g_2})$ | (2.0, 0.15) |
| | third goal configuration $(h_{g_3}, \sigma_{g_3})$ | (0.5, 0.2) |
| | fourth goal configuration $(h_{g_4}, \sigma_{g_4})$ | (0.25, 0.25) |
| | diffusion steps $N$ | 20 |
| | fast diffusion steps $N_f$ | 4 |

# E   Maze2D Gold-picking Test Map

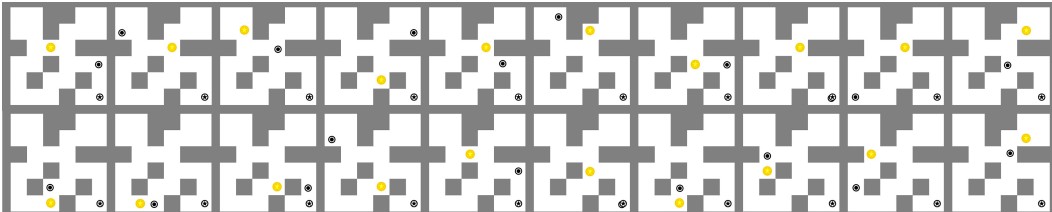

Figure 8: **Maze2d-Medium Single-Task Test Map.**

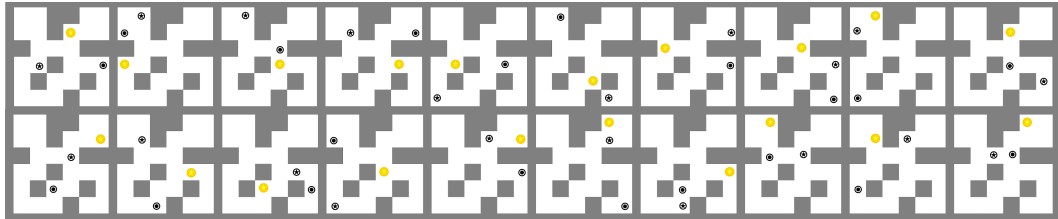

Figure 9: **Maze2d-Medium Multi-Task Test Map.**

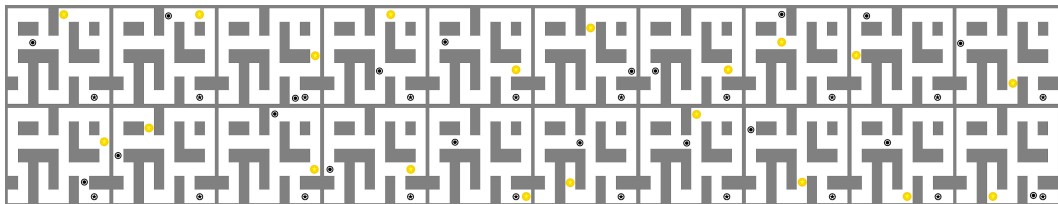

Figure 10: **Maze2d-Large Single-Task Test Map.**

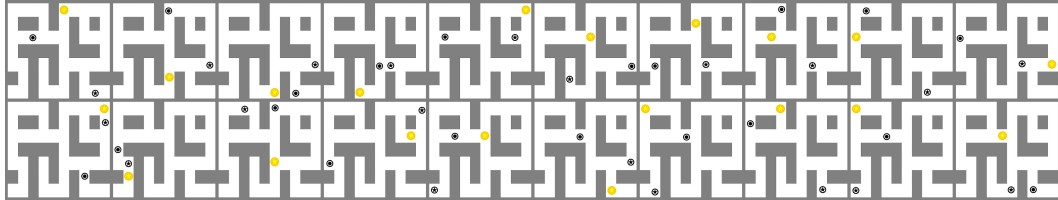

Figure 11: **Maze2d-Large Multi-Task Test Map.**

# F   Evaluation Metrics

We explain the evaluation metrics used in Table 2 and Table 3.

**KUKA Robot Arm Manipulation.**   In PNP (*stack*), the agent gets 1 point whenever it stacks a block on the existing block. The maximum points the agent can get is 3 points since there are 3 blocks to stack (one is fixed). In PNP (*place*), the agent gets 1 point whenever it moves a block to a desired space. The maximum points the agent can get is 4 points since there are 4 blocks to move. In PNWP, if the agent moves a block to a region of global optimum, it will get 1 point, while it can get only half a point if it moves a block to a region of local optimum. Therefore, the maximum score that the agent can achieve in this task is 4 points, as it is guided to move 4 blocks in total.

**AntMaze Multi-goal Exploration.**   The number of found goals denotes the number of goals that the agent has found until the task is ended. The number of timesteps per goal indicates the timesteps consumed to reach each goal. Sequence match evaluates how closely an agent follows a predefined

sequence of visits. For example, if the defined sequence is $1 \to 2 \to 3 \to 4$, the agent earns points based on the number of ordered pairs (*i.e.*, $\{(1,2),(1,3),(1,4),(2,3),(2,4),(3,4)\}$) included in its path. The maximum score an agent can obtain is 6 points ($\because \binom{4}{2} = 6$). If the agent's visiting order is $2 \to 3 \to 4 \to 1$, then the ordered pairs (2,3), (3,4), and (2,4) match the order in the defined sequence, resulting in a score of 3 points.

# G   Time Budget Analysis

We analyze the extra time cost of our method 1) to calculate pairwise distance of trajectory samples for particle guidance and 2) to expand sub-tree trajectories by fast-denoising, as shown in Table 5.

Table 5: **Planning Time Comparison in KUKA Robot Arm Manipulation.** The results are averaged over 100 planning seeds per trajectory generation. All test results are measured on a single NVIDIA GeForce RTX 3090 GPU core. The unit is second (s).

| Task | Diffuser | Diffuser$^\gamma$ | MCSS | TDP (w/o child) | TDP (w/o PG) | TDP |
|---|---|---|---|---|---|---|
| PNP (*stack·place*) | 16.28 | 16.73 | 16.39 | 17.25 | 18.02 | 19.08 |
| PNWP | 16.66 | 17.53 | 16.89 | 17.59 | 17.87 | 18.31 |

# H   Hyperparameter Selection

TDP relies on three key hyperparameters: gradient guidance scale $\alpha_g$, particle guidance scale $\alpha_p$, and fast denoising steps $N_f$. We provide an empirical analysis of these hyperparameters on the KUKA tasks (PNP (*place*) and PNWP), demonstrating the robustness of TDP across a broad range of values for each hyperparameter. Each result used 18 samples per planning. Bolded values indicate the defaults used in the main experiments, and the rest of the hyperparameters are fixed to their default values as denoted in Appendix D.

Table 6: **Effect of $N_f$ on TDP Performance.** $\pm$ denotes standard error.

| $N_f$ | 50 | **100** | 200 | 400 |
|---|---|---|---|---|
| PNP (*place*) | $37.1 \pm 0.08$ | $37.8 \pm 0.09$ | $38.1 \pm 0.08$ | $38.0 \pm 0.07$ |
| PNWP | $65.7 \pm 0.14$ | $67.1 \pm 0.10$ | $68.3 \pm 0.09$ | $68.8 \pm 0.10$ |

Table 7: **Effect of $\alpha_p$ on TDP Performance.** $\pm$ denotes standard error.

| $\alpha_p$ | 0.1 | 0.25 | **0.5** | 1.0 |
|---|---|---|---|---|
| PNP (*place*) | $37.4 \pm 0.08$ | $36.6 \pm 0.08$ | $37.8 \pm 0.09$ | $35.5 \pm 0.11$ |
| PNWP | $66.5 \pm 0.14$ | $66.1 \pm 0.11$ | $67.1 \pm 0.10$ | $64.8 \pm 0.13$ |

Table 8: **Effect of $\alpha_g$ on TDP Performance.** $\pm$ denotes standard error.

| $\alpha_g$ | 25 | 50 | **100** | 200 |
|---|---|---|---|---|
| PNP (*place*) | $38.1 \pm 0.08$ | $38.6 \pm 0.07$ | $37.8 \pm 0.09$ | $36.0 \pm 0.12$ |
| PNWP | $67.8 \pm 0.09$ | $68.2 \pm 0.11$ | $67.1 \pm 0.10$ | $64.0 \pm 0.14$ |

Based on the ablation study, we provide practical guidelines for selecting each hyperparameter for unseen test tasks.

- $N_f$: This can be selected based on the environment before planning time. A common heuristic is to set $N_f$ between 10% and 20% of the original diffusion steps used by the pretrained diffusion planner.
- $\alpha_p$: In practice, $\alpha_p$ typically lies within the range [0.1, 0.5]. This range is compatible across standard environments, where state spaces are normalized (*e.g.*, to [-1, 1]). Increasing $\alpha_p$ encourages greater diversity in parent trajectory sampling.
- $\alpha_g$: This should be empirically tuned based on the characteristics of the test-time task and guide function.

# I Ablation Study: Number of Samples

Our framework aggregates multiple trajectory samples during test-time planning, where the number of samples significantly impacts performance. While the main paper reports averaged results across different sampling budgets, here we provide a detailed ablation to show how performance varies with the number of samples.

## I.1 Kuka Robot Arm Manipulation

Table 9: **Result of PNP (*stack*).** $\pm$ denotes standard error.

| # samples | Diffuser$^{\gamma}$ | MCSS | TDP (w/o child) | TDP (w/o PG) | TDP |
|---|---|---|---|---|---|
| 6 | $59.3_{\pm 0.10}$ | $61.3_{\pm 0.10}$ | $62.0_{\pm 0.08}$ | $56.0_{\pm 0.10}$ | $62.3_{\pm 0.09}$ |
| 12 | $61.0_{\pm 0.09}$ | $58.7_{\pm 0.09}$ | $57.0_{\pm 0.09}$ | $59.7_{\pm 0.10}$ | $56.0_{\pm 0.09}$ |
| 18 | $58.7_{\pm 0.09}$ | $58.0_{\pm 0.10}$ | $62.0_{\pm 0.09}$ | $59.7_{\pm 0.09}$ | $66.0_{\pm 0.10}$ |
| 24 | $61.3_{\pm 0.08}$ | $61.7_{\pm 0.09}$ | $59.0_{\pm 0.10}$ | $62.3_{\pm 0.10}$ | $60.3_{\pm 0.09}$ |

Table 10: **Result of PNP (*place*).** $\pm$ denotes standard error.

| # samples | Diffuser$^{\gamma}$ | MCSS | TDP (w/o child) | TDP (w/o PG) | TDP |
|---|---|---|---|---|---|
| 6 | $20.8_{\pm 0.08}$ | $30.2_{\pm 0.08}$ | $31.8_{\pm 0.08}$ | $34.0_{\pm 0.08}$ | $35.2_{\pm 0.07}$ |
| 12 | $21.0_{\pm 0.07}$ | $31.2_{\pm 0.08}$ | $30.0_{\pm 0.08}$ | $37.2_{\pm 0.08}$ | $35.5_{\pm 0.08}$ |
| 18 | $21.2_{\pm 0.06}$ | $32.2_{\pm 0.07}$ | $33.2_{\pm 0.07}$ | $37.5_{\pm 0.07}$ | $37.8_{\pm 0.09}$ |
| 24 | $22.8_{\pm 0.07}$ | $31.8_{\pm 0.08}$ | $32.8_{\pm 0.07}$ | $39.0_{\pm 0.09}$ | $39.2_{\pm 0.08}$ |

Table 11: **Result of PNWP.** $\pm$ denotes standard error.

| # samples | Diffuser$^{\gamma}$ | MCSS | TDP (w/o child) | TDP (w/o PG) | TDP |
|---|---|---|---|---|---|
| 6 | $33.5_{\pm 0.05}$ | $34.8_{\pm 0.05}$ | $34.4_{\pm 0.05}$ | $64.5_{\pm 0.11}$ | $65.1_{\pm 0.11}$ |
| 12 | $33.6_{\pm 0.05}$ | $35.2_{\pm 0.05}$ | $36.0_{\pm 0.05}$ | $65.6_{\pm 0.10}$ | $65.0_{\pm 0.11}$ |
| 18 | $36.0_{\pm 0.05}$ | $35.5_{\pm 0.05}$ | $34.9_{\pm 0.05}$ | $67.6_{\pm 0.10}$ | $67.1_{\pm 0.10}$ |
| 24 | $35.8_{\pm 0.05}$ | $37.2_{\pm 0.04}$ | $36.9_{\pm 0.04}$ | $68.8_{\pm 0.10}$ | $70.0_{\pm 0.10}$ |

## I.2 AntMaze Multi-goal Exploration

Table 12: **Number of Found Goals** $\pm$ denotes standard error.

| # samples | Diffuser$^{\gamma}$ | MCSS | TDP (w/o child) | TDP (w/o PG) | TDP |
|---|---|---|---|---|---|
| 32 | $11.5_{\pm 1.6}$ | $53.5_{\pm 3.7}$ | $53.8_{\pm 3.8}$ | $59.8_{\pm 3.7}$ | $57.5_{\pm 3.9}$ |
| 64 | $13.5_{\pm 1.7}$ | $56.5_{\pm 4.1}$ | $73.8_{\pm 3.4}$ | $61.0_{\pm 3.8}$ | $73.8_{\pm 3.4}$ |
| 128 | $13.5_{\pm 1.9}$ | $66.0_{\pm 3.8}$ | $67.3_{\pm 3.7}$ | $67.3_{\pm 3.6}$ | $63.5_{\pm 3.6}$ |
| 256 | $11.3_{\pm 1.6}$ | $69.3_{\pm 3.9}$ | $70.8_{\pm 3.6}$ | $70.3_{\pm 3.9}$ | $69.5_{\pm 3.8}$ |

Table 13: **Sequence Match Score.** $\pm$ denotes standard error.

| # samples | Diffuser$^{\gamma}$ | MCSS | TDP (w/o child) | TDP (w/o PG) | TDP |
|---|---|---|---|---|---|
| 32 | $0.8_{\pm 0.22}$ | $25.3_{\pm 1.47}$ | $24.3_{\pm 1.59}$ | $27.7_{\pm 1.50}$ | $27.3_{\pm 1.57}$ |
| 64 | $1.2_{\pm 0.29}$ | $27.8_{\pm 1.68}$ | $35.0_{\pm 1.81}$ | $30.5_{\pm 1.70}$ | $36.5_{\pm 1.45}$ |
| 128 | $2.0_{\pm 0.47}$ | $33.7_{\pm 1.71}$ | $38.5_{\pm 1.74}$ | $34.8_{\pm 1.62}$ | $33.8_{\pm 1.87}$ |
| 256 | $0.8_{\pm 0.22}$ | $34.0_{\pm 1.52}$ | $37.3_{\pm 1.86}$ | $37.8_{\pm 1.78}$ | $37.5_{\pm 1.77}$ |

Table 14: **Number of timesteps per goal.** $\pm$ denotes standard error.

| # samples | Diffuser$^{\gamma}$ | MCSS | TDP (w/o child) | TDP (w/o PG) | TDP |
|---|---|---|---|---|---|
| 32 | 4347.8 | 801.1 | 773.4 | 661.6 | 708.4 |
| 64 | 3703.7 | 667.5 | 598.7 | 626.7 | 462.0 |
| 128 | 3703.7 | 549.0 | 545.2 | 564.3 | 594.1 |
| 256 | 4444.5 | 481.1 | 467.4 | 478.7 | 504.0 |

## J  TDP performance on the standard maze benchmark

TDP surpasses sequential approaches (*i.e.*, MCTD [11], Diffusion-Forcing [10]) on the standard maze offline benchmarks [51].

- pointmaze-{*medium*, *large*, *giant*}-navigate-v0: $100 \pm 0$
- antmaze-{*medium*, *large*}-navigate-v0: $100 \pm 0$
- antmaze-*giant*-navigate-v0: $98 \pm 6$

TDP has advantages over sequential approaches even in single-task scenarios, in terms of **trajectory optimality**. For instance, in standard maze benchmarks, the *shortest-path* trajectory between initial and goal states represents the globally optimal solution, while longer trajectories correspond to suboptimal (local optimal) solutions [4]. As maze environments scale up, identifying globally optimal solutions requires evaluating combinatorially diverse trajectory candidates with varying structures.

Single-step exploration methods like MCTD optimize actions one step at a time (via guided vs. non-guided action selection), which tends to lead to locally greedy decisions [52] and results in longer trajectories than the shortest path. In contrast, TDP employs a multi-step exploration framework via bi-level search, enabling per-step action selection based on multi-step future rewards evaluated across diverse-branched trajectory candidates. This approach allows TDP to be able to assess long-term action consequences and avoid the local optimal path.

## K  Learned PG vs. Fixed PG

Learned potentials [46] offer slight improvements in both sample diversity and overall performance. The results below compare TDP using learned versus fixed potentials on the KUKA benchmarks.

- **pairwise trajectory distance**: Learned PG ($17.54 _{\pm 0.04}$) vs. Fixed PG ($17.38 _{\pm 0.05}$)
- **performance**:
  - PNP (*stack*): Learned PG ($61.28 _{\pm 0.22}$) vs. Fixed PG ($61.17 _{\pm 0.24}$)
  - PNP (*place*): Learned PG ($37.04 _{\pm 0.12}$) vs. Fixed PG ($36.94 _{\pm 0.13}$)
  - PNWP: Learned PG ($67.23 _{\pm 0.15}$) vs. Fixed PG ($66.81 _{\pm 0.17}$)

However, training a task-specific particle guidance potential model requires expert demonstrations, making it impractical for unseen tasks.

