# OpenReview forum: "Tree-Guided Diffusion Planner"
_NeurIPS.cc/2025/Conference — NeurIPS 2025 poster_

### Official Review · Reviewer_2mBS · 2025-06-28

**Clarity:** 2
**Significance:** 3
**Originality:** 3
**Rating:** 4
**Confidence:** 4

**Summary:**

The authors tackle the limitations of standard gradient-based guidance in diffusion models. The authors identify that existing guidance methods often fail in scenarios with non-convex or non-differentiable objectives, which are common in realistic planning tasks, so the authors introduce the the Tree-guided Diffusion Planner (TDP), a novel training-free, test-time planning framework designed to overcome this limitation. TDP addresses this by structuring trajectory generation as a tree search that explicitly balances exploration and exploitation. It uses a bi-level sampling process: (1) diverse parent trajectories are generated using Particle Guidance to ensure broad exploration of the solution space, and (2) local sub-trajectories are refined from these parents using fast, gradient-guided denoising to exploit promising regions.

**Questions:**

1. See the weaknesses part. Is there anything I misunderstood or overlooked about the weakness I mentioned?
2. How much tuning effort is required to find effective hyperparameters values for a new task?
3. For tasks like Maze2D Gold-picking, a straightforward alternative would be to plan sequentially: first plan a trajectory to the intermediate goal (the gold), and then re-plan from that state to the final destination. This approach would also handle the non-differentiable constraint. Could you elaborate on the specific advantages of TDP's holistic, diffusion-based planning over such a simpler, sequential approach?

**Ethical Concerns:**

["NO or VERY MINOR ethics concerns only"]

**Final Justification:**

While the authors have addressed some of my concerns, particularly regarding the model's robustness to hyperparameters, several critical issues remain unresolved for me.

(1) I am still not fully convinced by the explanation for why TDP performs better than methods like MCTD, especially in single-task scenarios. The rebuttal did not provide a clear and compelling justification for the source of TDP's advantage in these single task settings (see in the discussion with Reviewer GPcL).

(2) I also question the paper's reproducibility, as the code for TDP has not been made available and many implementation details are missing from the paper. Therefore, I am not confident that I could fully reproduce the results on my own.

--- update ---
With the authors’ further clarifications, most of my concerns have been well addressed, particularly regarding the contribution compared to MCTD and the reproducibility issue. Therefore, I have raised my score to 4.

**Limitations:**

yes

**Paper Formatting Concerns:**

No major formatting issues were found.

**Quality:**

3

**Strengths And Weaknesses:**

Strengths:
1. The paper is well-motivated. It clearly presents a key weakness of current diffusion planners: the reliance on simple, gradient-based guidance that is ill-suited for the complex, non-convex, and non-differentiable objective functions often encountered in the real world.
2. The paper is clearly written, with useful illustrations.
3. The paper provides theoretical justification of bi-level sampling process.

Weaknesses:
1. Although the method is training-free, it introduces a new set of important test-time hyperparameters. While the paper commendably provides a detailed ablation study on the number of samples, it lacks a similar sensitivity analysis for other critical test-time hyperparameters introduced by the method. Specifically, the guidance strengths for particle guidance, and gradient guidance, as well as the number of fast denoising steps. Without an analysis of how robust the method is to changes in these parameters, it is unclear how much tuning effort is required for a new task. This potentially replaces a "training" burden with a significant "test-time tuning" burden for these specific parameters."
2. A major weakness of this paper is the lack of comparison and discussion of highly relevant works. In particular, [1] introduces stochastic sampling, an MCMC-based method, alternative sampling-based solution to the limitations of standard gradient guidance that TDP also aims to solve. Similarly, [2] introduces Monte Carlo Tree Diffusion (MCTD) which also employs a tree-based search to guide the diffusion process at test time. It uses meta-actions (e.g., guide / no guide) to navigate the exploration-exploitation trade-ff at the level of sub-plans. Given the absence of a compartive discussion makes it difficult to assess the unique contributions of TDP.

[1] Inference-Time Policy Steering through Human Interactions

[2] Monte Carlo Tree Diffusion for System 2 Planning

---

> ### Author Rebuttal · Authors · 2025-07-31
>
> We appreciate Reviewer **2mBS** for the detailed and thoughtful suggestions.
> > Is there anything I misunderstood or overlooked about the weakness I mentioned?
>
> While most of your observations are valid, we believe there are some aspects regarding test-time tuning burden (Weakness 1) and sequential planning approach (Question 3) that may benefit from additional clarification. We will address all the concerns you raised below.
>
> > A major weakness of this paper is the lack of comparison and discussion of highly relevant works.
>
> We agree it is important to clearly position TDP and articulate how it differs from the relevant approaches.
>
> **1. TDP vs. Existing Approaches**
>
> We would like to clarify why existing methods are not well-suited to solve our tested benchmarks. We provide three case studies to elaborate the difference between TDP and the existing methods.
>
> **1-1. Sequential approaches** struggle with multi-goal benchmarks such as PnWP and AntMaze Multi-goal Exploration. For example, MCTD [b] performs **single-step** exploration by selecting between guided and unguided actions at each time step. This works well on standard offline RL benchmarks with convex structures and a single optimal goal (e.g., offline maze navigation). However, per-action exploration causes MCTD to often fail discovering distant high-reward goals and instead converges to locally optimal solutions. The PnWP benchmark captures this limitation, and the AntMaze Multi-goal task requires identifying better goals to maximize sequence match scores. In contrast, TDP performs **multi-step** exploration through diverse bi-level trajectory sampling, better handling challenging multi-goal scenarios.
>
> **1-2. Hierarchical diffusion planners** improve compositional behaviors using subgoals and intermediate rewards [31, 4]. Hierarchical methods excel in generalizing over **labeled** data distributions seen during training. Our benchmark suites consist entirely of **unlabeled** tasks unseen during training, posing a challenge for hierarchical planners that rely on training-time supervision. For example, while prior work [4] demonstrates that hierarchical diffusion planners can generate *shortest-path* trajectories when both initial and goal states are specified, our gold-picking benchmark presents a fundamentally different challenge: the goal is *hidden* and often not aligned with the shortest path.
>
> **1-3. Diffusion model predictive control** (D-MPC) [d] learns multi-step proposals using dynamics models $p(s|a)$ to enable adaptation to test-time changes in system dynamics. However, it struggles to generalize to **unseen long-horizon** tasks, as modeling behavior solely through $p(s|a)$ can be limiting. Standard dynamics models struggle to capture long-context reward structures effectively. In contrast, TDP models the joint distribution $p(s,a)$, effectively solving long-horizon and multi-goal tasks. Furthermore, D-MPC is a **few-shot** learner, requiring fine-tuning with a small set of expert demonstrations. TDP is a fully zero-shot planner, operating without test-time demonstrations.
>
> **2. Supervised Planning vs. Zero-shot Planning**
>
> Most of the recent diffusion planner works focus on enhancing **supervised** planning capabilities [a, b, 32, 31, 4, d]. They are categorized into sequential approaches [a, b], hierarchical approaches [31, 4], and fine-tuning approaches [32, d]. These works improve the modeling of underlying system dynamics from the static offline RL benchmark. The recent works successfully solve challenging offline benchmarks via reformulating the training scheme, learning value estimator, and test-time scaling.
> Another line of recent diffusion planner research focuses on enhancing **zero-shot** planning capabilities [c]. The test-time tasks are shifted from the trained distribution, and planners are given access to a pretrained model (i.e., Diffuser) along with dense reward signals to effectively adapt to these unseen tasks. **TDP** is a task-aware planning framework, exhibiting robust zero-shot planning performance in a variety of tasks.
>
> **3. Additional baseline**
>
> We added a baseline the reviewer has mentioned, **MCSS+SS**, for all test tasks. MCSS+SS denotes Stochastic Sampling [c] with best-of-batch selection based on the guide score.
> - **MCSS+SS vs. TDP**
>   - Maze2D-*Medium*/*Large*: MCSS+SS (17.4±3.2 / 21.2±3.5) vs. TDP (**39.8±4.2 / 47.6±4.1**)
>   - Multi2D-*Medium*/*Large*: MCSS+SS (29.2±3.6 / 58.0±3.8) vs. TDP (**74.7±3.0 / 70.0±3.5**)
>   - PnP (stack) / PnP (place) / PnWP: MCSS+SS (56.8±0.1 / 35.5±0.19 / 36.24±0.09) vs. TDP (**61.17±0.24 / 36.94±0.13 / 66.81±0.17**)
>   - AntMaze Multi-goal: # goals (MCSS+SS: 62.8 vs. **TDP: 66.1**), seq. match (31.2 vs. **33.8**), timesteps (604.4 vs. **558.4**)
>
> > Could you elaborate on the specific advantages of TDP's holistic, diffusion-based planning over such a simpler, sequential approach?
>
> We clarify that the gold-picking task is a **black-box** problem that requires inferring the intermediate goal (i.e., gold) location using only a distance-based guide function, without access to the gold’s exact position. Contrary to the reviewer’s suggestion, it is not feasible to plan two separate trajectories sequentially, as the intermediate goal position is not explicitly known. To address this, TDP performs bi-level trajectory sampling to broadly explore the environment and collect guide signals, enabling it to discover the *hidden* gold location within the map.
>
> > Without an analysis of how robust the method is to changes in these parameters, it is unclear how much tuning effort is required for a new task.
>
> We agree that the robustness of TDP with respect to the variety of test-time hyperparameters should be elaborated since TDP aims to solve unseen tasks in an efficient manner.
> TDP relies on three test-time hyperparameters, as noted by the reviewer: gradient guidance scale $\alpha_g$, particle guidance scale $\alpha_p$, and fast denoising steps $N_f$. We provide an empirical analysis of these hyperparameters on the Kuka tasks (PnP (*place*) and PnWP), demonstrating the **robustness** of TDP across a broad range of values for each hyperparameter. Each result used 18 samples per planning. Bolded values indicate the default hyperparameter values and their corresponding results in the main experiments, and the rest of hyperparameters are fixed to their default values.
> - **$N_f$**: 50 / **100** / 200 / 400
>    - PnP (*place*): 37.1 ± 0.08 / **37.8 ± 0.09** / 38.1 ± 0.08 / 38.0 ± 0.07
>    - PnWP: 65.7 ± 0.14 / **67.1 ± 0.10** / 68.3 ± 0.09 / 68.8 ± 0.10
>
> - **$\alpha_p$**: 0.1 / 0.25 / **0.5** / 1.0
>    - PnP (*place*): 37.4 ± 0.08 / 36.6 ± 0.08 / **37.8 ± 0.09** / 35.5 ± 0.11
>    - PnWP: 66.5 ± 0.14 / 66.1 ± 0.11 / **67.1 ± 0.10** / 64.8 ± 0.13
>
> - **$\alpha_g$**: 25 / 50 / **100** / 200
>    - PnP (*place*): 38.1 ± 0.08 / 38.6 ± 0.07 / **37.8 ± 0.09** / 36.0 ± 0.12
>    - PnWP: 67.8 ± 0.09 / 68.2 ± 0.11 / **67.1± 0.10** / 64.0 ± 0.14
>
> In addition, we provide practical guidelines for selecting each value as follows.
> - **$N_f$**: It can be selected based on the environment before planning time. A common heuristic is to set $N_f$ between 10% and 20% of the original diffusion steps used by the pretrained diffusion planner.
> - **$\alpha_p$**: In practice, $\alpha_p$ typically lies within the range [0.1, 0.5]. This range is compatible across standard environments, where state spaces are normalized (e.g. to [-1, 1]). Increasing $\alpha_p$ encourages greater diversity in parent trajectory sampling.
> - **$\alpha_g$**: It should be empirically tuned based on the characteristics of the test-time task and guide function.
>
> > This potentially replaces a "training" burden with a significant "test-time tuning" burden for these specific parameters.
>
> We would like to clarify why TDP can be an alternative way to solve test-time task rather than training-per-task approaches. TDP’s test-time scalability and generalization capability enable **zero-shot** planning that surpasses fine-tuning approaches in both **performance** and **time cost**. As a standard baseline, **AdaptDiffuser** [32] refers to the fine-tuned diffuser model on synthetic expert demonstrations. TDP outperforms AdaptDiffuser on both standard pick-and-place benchmarks (PnP) and the custom task (PnWP).
> - **Performance**
> |**Task**|**AdaptDiffuser**|**TDP**|
> |:-|-|-|
> |PnP (*stack*)|60.54 ± 0.18|**61.17** ± 0.24|
> |PnP (*place*)|36.17 ± 0.11|**36.94** ± 0.13|
> |PnWP|39.72 ± 0.08|**66.81**± 0.17|
>
> - **Time cost**  (on NVIDIA RTX 3090)
>    - **AdaptDiffuser**: 48 hours (collecting synthetic expert demonstrations) + 2 hours (fine-tuning)
>    - **TDP**: <1 hour (test-time hyperparameter tuning based on the practical guideline above)
>
> Furthermore, collecting expert demonstrations for each test-time task is often impractical, especially in tasks with complex dynamics or limited simulation control. It can be challenging to extract a sufficient number of high-quality trajectories on demand at test time. In contrast, TDP offers a plug-and-play solution that operates directly on top of pretrained diffusion planners—requiring no additional supervision or adaptation—making it a scalable and efficient alternative for diverse unseen tasks.
>
> > How much tuning effort is required to find effective hyperparameters values for a new task?
>
> The values of $\alpha_p$ and $N_f$ typically generalize well across tasks within the same environment. In contrast, $\alpha_g$ is more task-specific and requires heuristic tuning. However, this tuning remains substantially more efficient—both in time and computation—than fine-tuning approaches.
>
> [a] Boyuan Chen et al. Diffusion Forcing: Next-token Prediction Meets Full-Sequence Diffusion. 2024.
> [b] Jaesik Yoon et al. Monte Carlo Tree Diffusion for System 2 Planning. 2025.
> [c] Yanwei Wang et al. Inference-Time Policy Steering through Human Interactions. 2025.
> [d] Guangyao Zhou et al. Diffusion Model Predictive Control. 2025.

---

> > ### Comment · Reviewer_2mBS · 2025-08-05
> >
> > Thank for detailed reponse or addressing some of my concerns. I have read your response carefully, however, a few points remain unclear for me.
> >
> > For instance, I am not fully convinced by the explanation of why TDP outperforms a method like MCTD, even in the single-task scenarios mentioned in the discussion with Reviewer GPcL. I think more explanations and evidences should be provided. I also still question about the paper's reproducibility, as the code for TDP has not been made available and many implementation details are missing from the paper. Therefore, I am not confident that I could fully reproduce the results on my own.
> >
> > I decided to maintain my initial borderline reject rating, leaning more toward borderline, although I would not oppose the paper's acceptance.

---

> > > ### Author Response · Authors · 2025-08-05
> > >
> > > Thank you for your detailed review and for carefully considering our responses. We appreciate your thoughtful engagement and would like to address your remaining concerns point by point.
> > >
> > > > I am not fully convinced by the explanation of why TDP outperforms a method like MCTD, even in the single-task scenarios mentioned in the discussion with Reviewer GPcL.
> > >
> > > We would like to clarify that TDP has advantages over MCTD even in single-task scenarios, in terms of **trajectory optimality**. For instance, in standard maze benchmarks, the *shortest-path* trajectory between initial and goal states represents the globally optimal solution, while longer trajectories correspond to suboptimal (local optimal) solutions [4]. As maze environments scale up, identifying globally optimal solutions requires evaluating combinatorially diverse trajectory candidates with varying structures.
> > >
> > > **Single-step** exploration methods like MCTD optimize actions one step at a time (via guided vs. non-guided action selection), which tends to lead to **locally greedy decisions** [b] and results in longer trajectories than the shortest path (see Figure 2 in [a]).
> > > In contrast, TDP employs a **multi-step** exploration framework via bi-level search, enabling per-step action selection based on multi-step future rewards evaluated across **diverse-branched** trajectory candidates. This approach allows TDP to be able to assess long-term action consequences and avoid the local optimal path.
> > >
> > > > I also still question about the paper's reproducibility, as the code for TDP has not been made available and many implementation details are missing from the paper.
> > >
> > > We appreciate your concern regarding reproducibility. We are happy to release the complete code for TDP upon paper acceptance. In the current submission, we have provided comprehensive implementation details through step-by-step guidelines in Algorithm 1, 2, and 3, which outline TDP’s complete pipeline. Additionally, all hyperparameters for every experiment reported in the main paper are documented in Appendix C, ensuring full reproducibility of our results.
> > >
> > > We believe these algorithmic descriptions and parameter specifications provide sufficient detail for reproduction, and the code release upon acceptance will further enable researchers to build upon our work. Additionally, we hope that our work can contribute to the community by providing comprehensive zero-shot planning benchmarks.
> > >
> > > [a] Jaesik Yoon et al. Monte Carlo Tree Diffusion for System 2 Planning. 2025.
> > > [b] C. B. Browne et al. A Survey of Monte Carlo Tree Search Method. 2012.

---

> > > > ### Author Response · Authors · 2025-08-08
> > > >
> > > > Dear Reviewer **2mBS**,
> > > >
> > > > Thank you again for your detailed review and for considering our rebuttal. We sincerely appreciate your openness to acceptance and the careful attention you’ve given to evaluating our work.
> > > >
> > > > In our earlier response, we aimed to address your remaining concerns — particularly the comparisons with MCTD and reproducibility.
> > > >
> > > > We would be glad to know if these clarifications are sufficient. We value your thoughtful input and engagement.
> > > >
> > > > If any concerns still remain unresolved, we are more than happy to provide additional explanations, evidence, or implementation details to ensure clarity before the discussion period concludes.
> > > >
> > > > Sincerely,
> > > > The Authors of Submission #26284

---

> > > > > ### Comment · Reviewer_2mBS · 2025-08-08
> > > > >
> > > > > Thank you to the authors for the patient clarifications, which have addressed my concerns. I strongly encourage including the additional experimental results and all rebuttal clarifications in the revised paper. Given the commitment to release the complete code upon acceptance, I have raised my score to 4.

---

> > > > > > ### Author Response · Authors · 2025-08-08
> > > > > >
> > > > > > We sincerely appreciate your thoughtful engagement and for increasing your score. We are grateful for your acknowledgment that our clarifications have addressed your concerns. We will ensure that the additional experimental results and all rebuttal clarifications are incorporated into the revised version. We remain committed to releasing the complete code upon acceptance so the community can fully reproduce and build upon our work.

---

### Official Review · Reviewer_GPcL · 2025-07-02

**Clarity:** 3
**Significance:** 3
**Originality:** 3
**Rating:** 4
**Confidence:** 4

**Summary:**

The authors propose TDP, a test-time planning framework of diffusion planner for offline RL tasks. TDP consists of two stages: Parent Branching and Sub-tree Expansion. This bi-level framework effectively balance exploration-exploitation trade-off.

**Questions:**

Here are some questions I want to ask:

- It seems that the computational cost of TDP is similar to a naive Diffuser, even though it has a test-time adaptation stage. Could authors explain how is it possible? Furthermore, I would like to ask if we have more computational budget for test-time, would the performance improve?
- Decomposing states into observation and control states is possible only when the state is given as a compact vector. What if the state is given as a visual input?

**Ethical Concerns:**

["NO or VERY MINOR ethics concerns only"]

**Final Justification:**

Update the score based on rebuttal response: Discussion on relevant works / Competitive performance on additional benchmarks.

**Limitations:**

Here are some comments I want to suggest:

- As written in the weakness part, there are several works that consider test-time planning of diffusers for improving performance. I strongly recommend authors to conduct more comprehensive literature review and add a discussion part for these works.
- It might be better to add some analysis on hyperparameters. It seems that we use different guidance scale for different guidance mechanisms and environments. It would be also nice to add some studies related on quality of planning and time complexity in terms of $N_f$

**Paper Formatting Concerns:**

I do not notice any formatting issues.

**Quality:**

3

**Strengths And Weaknesses:**

**Strengths**

- The authors consider an important topic, test-time scaling of diffusion planners.
- Extensive experiments results across diverse tasks
- Code is available in the supplementary material

**Weakness**

- Test-time scaling of diffusion planners is an important topic in offline RL tasks and there are several recent works, which also utilize tree-like structure [1-3]. I strongly recommend authors to compare their method with these works or at least add a discussion.
- While I acknowledge that authors conduct experiments on diverse benchmarks such as Kuka robot arm manipulation, it would be nice to conduct experiments on much larger tasks such as pointmaze and antmaze giant, proposed by OGBench. Experiment results on those tasks is beneficial for the authors’ claim: bi-level approach for test-time planning is crucial.

[1] Yoon, Jaesik, et al. "Monte Carlo Tree Diffusion for System 2 Planning." *arXiv preprint arXiv:2502.07202* (2025).

[2] Zhou, Guangyao, et al. "Diffusion model predictive control." *arXiv preprint arXiv:2410.05364* (2024).

[3] Chen, Boyuan, et al. "Diffusion forcing: Next-token prediction meets full-sequence diffusion." *Advances in Neural Information Processing Systems* 37 (2024): 24081-24125.

---

> ### Author Rebuttal · Authors · 2025-07-30
>
> We sincerely thank Reviewer **GPcL** for the thoughtful suggestions. Below, we address each of the raised points.
>
> > ... I strongly recommend authors to compare their method with these works or at least add a discussion.
>
> We agree it is important to clearly position TDP and articulate how TDP differs from the recent approaches.
>
> **1. TDP vs. Existing Approaches**
> We would like to clarify why the existing tree-based methods show limited performance on our tested benchmarks. We provide three case studies to elaborate on the differences between TDP and existing methods.
>
> **1-1. Sequential approaches** struggle with multi-goal benchmarks such as PnWP and AntMaze Multi-goal Exploration. MCTD [b] performs **single-step** exploration by selecting between guided and unguided actions at each time step. This approach works well on standard offline RL benchmarks, where the tasks present convex structures with a single optimal goal (e.g., offline maze navigation). However, per-action exploration causes MCTD to often fail to discover distant high-reward goals, converging to local optima. The PnWP benchmark captures this limitation, and the AntMaze Multi-goal Exploration requires identifying better goals to maximize the sequence match score. In contrast, TDP performs **multi-step** exploration through diverse bi-level trajectory sampling, handling the challenging multi-goal scenarios better.
>
> **1-2. Hierarchical diffusion planners** improve compositional behaviors through the use of subgoals and intermediate rewards [31, 4]. Hierarchical methods excel in generalizing over **labeled** data distributions seen during training. Our tested benchmark suites consist entirely of **unlabeled** tasks that were not seen during training, posing a challenge for hierarchical planners relying on training-time supervision. While prior work [4] demonstrates that hierarchical diffusion planners can generate *shortest-path* trajectories when both initial and goal states are specified, our gold-picking benchmark presents a fundamentally different challenge: the goal is *hidden* and often not aligned with the shortest path.
>
> **1-3. Diffusion model predictive control** (D-MPC) [d] learns multi-step proposals using dynamics models $p(s|a)$ to enable adaptation to test-time changes in system dynamics. However, it struggles to generalize to **unseen long-horizon** tasks, as modeling such behavior solely through $p(s|a)$ can be limiting. For example, hovering or complex horizontal navigation remains difficult to capture with standard dynamics models. In contrast, TDP is a Diffuser-based planning framework that models joint distribution $p(s,a)$, effectively solving long-horizon and multi-goal tasks. Furthermore, D-MPC is a **few-shot** learner, requiring fine-tuning with a small set of expert demonstrations. TDP, on the other hand, is a fully zero-shot planner, operating without any additional demonstrations at test time.
>
> **2. Supervised Planning vs. Zero-shot Planning**
> Most of the recent diffusion planner works focus on enhancing **supervised** planning capabilities [a, b, 32, 31, 4, d]. They are categorized into sequential approaches [a, b], hierarchical approaches [31, 4], and fine-tuning approaches [32, d]. These works improve the modeling of underlying system dynamics from the static offline RL benchmarks. Recent works successfully solve challenging offline benchmarks via reformulating the training scheme, learning value estimator, and test-time scaling.
> Another line of recent diffusion planner research focuses on enhancing **zero-shot** planning capabilities [c]. Test-time tasks are shifted from the training distribution, and planners are given access to a pretrained model (i.e., Diffuser) along with dense reward signals to effectively adapt to these unseen tasks. **TDP** is a task-aware planning framework, exhibiting robust zero-shot planning performance in a variety of tasks.
>
> We will add this discussion to the main paper to better position TDP relative to existing approaches.
>
> **3. Additional Experiments**
> We provide the additional experiment results to compare TDP with the recent works. We briefly summarize three takeaways from our additional experiments below.
> - TDP surpasses sequential approaches (MCTD [b], Diffusion-Forcing [a]) on the standard maze offline benchmarks.
> - TDP surpasses the fine-tuning approach (AdaptDiffuser [32]) on both the standard pick-and-place benchmark (PnP) and the custom task (PnWP). It supports TDP’s test-time scalability and generalization capability.
> - TDP surpasses the recent zero-shot planning approach (MCSS+SS [c]) on all test tasks.
>
> **3-1. Result on the standard maze navigation benchmark**
>
> The results of Diffusion-Forcing and MCTD are referenced from [b].
> - PointMaze (Maze2D)
> |**Dataset**|**Diffusion-Forcing**|**MCTD**|**TDP**|
> |:-|-|-|-|
> |pointmaze-*medium*|65 ± 16|100 ± 0|**100** ± 0|
> |pointmaze-*large*|74 ± 9|98 ± 6|**100** ± 0|
> |pointmaze-*giant*|50 ± 10|100 ± 0|**100** ± 0|
>
> - AntMaze
> |**Dataset**|**Diffusion-Forcing**|**MCTD**|**TDP**|
> |:-|-|-|-|
> |antmaze-*medium*|90 ± 10|100 ± 0|**100** ± 0|
> |antmaze-*large*|57 ± 6|98 ± 6|**100** ± 0|
> |antmaze-*giant*|24 ± 12|94 ± 9|**98** ± 6|
>
> **3-2. Additional baselines on our tested benchmarks**
> We added two baselines: **AdaptDiffuser** for robot-arm manipulation and **MCSS+SS** for all test tasks. AdaptDiffuser is a fine-tuned diffusion planner on synthetic expert pick-and-place demonstrations, and MCSS+SS denotes Stochastic Sampling [c] with best-of-batch selection based on the guide score.
> - Maze2D gold-picking
> |**Environment**|**MCSS+SS**|**TDP**|
> |:-|-|-|
> |Maze2D-*Medium*|17.4±3.2|**39.8** ± 4.2|
> |Maze2D-*Large*|21.2±3.5|**47.6** ± 4.1|
> |Multi2D-*Medium*|29.2±3.6|**74.7** ± 3.0|
> |Multi2D-*Large*|58.0±3.8|**70.0** ± 3.5|
> - Pick-and-Place (PnP) and Pick-and-Where-to-Place (PnWP)
> |**Task**|**AdaptDiffuser**|**MCSS+SS**|**TDP**|
> |:-|-|-|-|
> |PnP (*stack*)|60.54 ± 0.18|56.8 ± 0.16|**61.17** ± 0.24|
> |PnP (*place*)|36.17 ± 0.11|35.5±0.19|**36.94** ± 0.13|
> |PnWP|39.72 ± 0.08|36.24 ± 0.09|**66.81**± 0.17|
> - AntMaze Multi-goal Exploration
> |**Metric**|**MCSS+SS**|**TDP**|
> |:-|-|-|
> |# found goals &uparrow;|62.8±1.9|**66.1** ± 1.9|
> |sequence match &uparrow;|31.2±1.3|**33.8** ± 1.4|
> |# timesteps per goal &downarrow;|604.4|**558.4**|
>
> > ... it would be nice to conduct experiments on much larger tasks such as pointmaze and antmaze giant, proposed by OGBench.
>
>  In our initial response, we provide the results on the pointmaze-*giant* and antmaze-*giant* standard navigation tasks, evaluated against MCTD and Diffusion-forcing. Our tested benchmarks are similarly long-horizon as OGBench’s standard maze tasks with a giant map. In the original datasets, the trajectory horizons are 1000 steps for medium and large maps, and 2000 steps for giant maps. Our multi-goal exploration task on a large map is designed with comparable long-context, requiring up to 2000 steps to complete.
>
> > It might be better to add some analysis on hyperparameters.
>
> TDP relies on three key hyperparameters as noted by the reviewer: gradient guidance scale $\alpha\_g$, particle guidance scale $\alpha\_p$, and fast denoising steps $N\_f$. We provide an empirical analysis of these hyperparameters on the Kuka tasks (PnP (*place*) and PnWP), demonstrating the **robustness** of TDP across a broad range of values for each hyperparameter. Each result used 18 samples per planning. Bolded values indicate the defaults used in the main experiments, and the rest of the hyperparameters are fixed to their default values.
> - $N\_f$: 50 / **100** / 200 / 400
>     - PnP (*place*): 37.1 ± 0.08 / **37.8 ± 0.09** / 38.1 ± 0.08 / 38.0 ± 0.07
>    - PnWP: 65.7 ± 0.14 / **67.1 ± 0.10** / 68.3 ± 0.09 / 68.8 ± 0.10
> - $\alpha\_p$: 0.1 / 0.25 / **0.5** / 1.0
>     - PnP (*place*): 37.4 ± 0.08 / 36.6 ± 0.08 / **37.8 ± 0.09** / 35.5 ± 0.11
>    - PnWP:  66.5± 0.14 / 66.1± 0.11 / **67.1± 0.10** / 64.8 ± 0.13
>  - $\alpha\_g$: 25 / 50 / **100** / 200
>    - PnP (*place*): 38.1 ± 0.08 / 38.6 ± 0.07 / **37.8 ± 0.09** / 36.0 ± 0.12
>    - PnWP: 67.8 ± 0.09 / 68.2 ± 0.11 / **67.1± 0.10** / 64.0 ± 0.14
>
> We provide practical guidelines for selecting each value as below.
> - $N\_f$: It can be selected based on the environment before planning time. A common heuristic is to set $N_f$ between 10-20% of the original diffusion steps used by the pretrained planner.
> - $\alpha\_p$: In practice, it typically lies within the range [0.1, 0.5]. This range is compatible across standard environments, where state spaces are normalized (e.g., to [-1, 1]). Increasing $\alpha_p$ encourages greater diversity in parent trajectory sampling.
> - $\alpha\_g$: It should be empirically tuned based on the characteristics of the test-time task and guide function.
>
> > ### Response to Q1 (Computational cost and scaling)
>
> TDP introduces only marginal computational overhead compared to the standard Diffuser, due to its use of fast-denoising in the Sub-Tree Expansion phase. For instance, in the Kuka environment, while the original diffusion process uses 1000 denoising steps, TDP generates child trajectories with just 100 steps, only 10% of the original cost. Also, we provide extensive experimental results in Appendix G, where we vary the number of samples per planning. This study shows that increasing test-time computational budget does not always guarantee better performance.
>
> > ### Response to Q2 (State decomposition for visual inputs)
>
> Visual inputs can be converted into *proxy* states with compact vector representations—by training inverse dynamics and position estimator models, following the approach used for visual maze in prior work [b].
>
> [a] Boyuan Chen et al. Diffusion Forcing: Next-token Prediction Meets Full-Sequence Diffusion. 2024.
> [b] Jaesik Yoon et al. Monte Carlo Tree Diffusion for System 2 Planning. 2025.
> [c] Yanwei Wang et al. Inference-Time Policy Steering through Human Interactions. 2025.
> [d] Guangyao Zhou et al. Diffusion Model Predictive Control. 2025.

---

> > ### Comment · Reviewer_GPcL · 2025-08-04
> >
> > Thanks for authors' response and conducting extensive experiments. I raised score to 4. I hope that discussion on relevant works and additional experiments are included in the final manuscript.

---

> > > ### Author Response · Authors · 2025-08-05
> > >
> > > We sincerely thank you for taking the time to provide your thoughtful feedback and for increasing the score to 4. We greatly appreciate your constructive feedback and will carefully address your suggestions regarding the discussion of relevant works and additional experiments in the final manuscript. Thank you again for your valuable guidance.

---

### Official Review · Reviewer_gnZK · 2025-07-02

**Clarity:** 2
**Significance:** 2
**Originality:** 2
**Rating:** 4
**Confidence:** 4

**Summary:**

The Tree-guided Diffusion Planner (TDP) introduces a training-free, bi-level trajectory-level tree search that first diversifies “parent” trajectories with particle guidance to explore broadly, and then refines each branch locally with fast gradient-guided denoising, balancing exploration and exploitation. Across Maze2D gold-picking, KUKA robot arm manipulation, and AntMaze multi-goal exploration tasks, TDP consistently surpasses diffusion-based planners.

**Questions:**

1. Definition of trajectory quality. Could you specify whether “quality” refers solely to guide reward J, dynamic feasibility, or another composite metric?

2. Learned vs fixed PG. The paper justifies fixed-potential PG for efficiency. Would a learned potential (as in [6]), paired with Sub-Tree Expansion, further improve convergence?

**Ethical Concerns:**

["NO or VERY MINOR ethics concerns only"]

**Final Justification:**

Through the authors' thorough rebuttal and discussion, all of my concerns have been fully addressed. I hope that the strengthened experimental results and clearer explanations will be reflected in the camera-ready version, and I am raising my score to a 4.

**Limitations:**

Yes

**Paper Formatting Concerns:**

No concerns about the paper formatting.

**Quality:**

2

**Strengths And Weaknesses:**

### Strengths
- **Theoretical analysis.** The paper proves (Proposition 1) that, under a two-Gaussian guide function and appropriate initialization, the bi-level sampling in TDP converges to the global optimum, whereas naïve gradient guidance only reaches a local one. Although the assumptions are stylised, giving any non-trivial optimality proof for diffusion-based planners is novel and valuable.

- **Training-free, test-time computation method.** TDP strengthens a pre-trained diffuser purely at inference time by (i) particle-guided exploration and (ii) gradient-guided local refinement, then picks the highest-scoring leaf trajectory. This compute-at-test-time paradigm is timely, mirroring the machine learning community’s shift toward fast task adaptation without retraining.

- **Empirical gains on diverse guidance shapes.** Across non-convex (Maze2D gold-picking), mixed dense-sparse reward setting (KUKA PNWP), and multi-goal (AntMaze) tasks, TDP surpasses both single-level samplers (Diffuser, MCSS) and tree-only baselines (TAT) by 10–25 %.

### Weaknesses
- **Narrow experimental coverage.** The evaluation is confined to three custom tasks. Because TDP is presented as a planner for long-horizon problems, it should also be benchmarked on the standard D4RL Maze2D/AntMaze suites and on OGBench [1], which offers a wide variety of goal-conditioned RL environments for competition with more varied and relatively new diffusion planner baselines.

- **Heuristic state decomposition.** The observation-control state split criteria should be handcrafted for each domain.  This manual partitioning requires substantial domain knowledge and may not scale to high-dimensional sensor spaces.

- **Ad-hoc dual guidance scheme.** Applying particle guidance to control states and gradient guidance to observation states departs from the principled formulation of conditional generation of classifier(-free) guidance, which is intended to model a single joint conditional distribution. This dichotomy feels heuristic, weakening the theoretical grounding of the method.

- **Insufficient methodological clarity.** Although Algorithm 1 appears in Appendix A, the main text does not provide a step-by-step description of how Parent Branching, Sub-Tree Expansion, leaf evaluation, and action execution interact. Readers must infer whether the planner operates in an open-loop fashion or replans after each action is executed.

- **Task-specific reward engineering.** TDP introduces a dense, hand-crafted reward 𝐽 for each environment. Because performance may be tightly coupled to how 𝐽 is shaped, TDP depends on domain-specific reward design and cannot be considered fully task-agnostic. Tasks in which such a distance is ill-defined or uninformative—e.g., OGBench’s AntSoccer or other high-dimensional, non-Euclidean goal spaces—would require substantial redesign, limiting the method’s generality and task-agnostic appeal.

[1] Park, Seohong, et al. "Ogbench: Benchmarking offline goal-conditioned rl." arXiv preprint arXiv:2410.20092 (2024).

---

> ### Author Rebuttal · Authors · 2025-07-29
>
> We sincerely appreciate Reviewer  **gnZK** for the detailed and constructive feedback. Below, we address each concern you raised and clarify the perceived weaknesses.
>
> > **Narrow experimental coverage** ...
>
> We provide more experiment results compared with the recent diffusion planner baselines: Diffusion-Forcing (2024) [a], MCTD (2025) [b], AdaptDiffuser (2023) [32], Stochastic Sampling (2025) [c]. We would like to briefly summarize three takeaways from our additional experiments first.
> - TDP surpasses sequential approaches (MCTD, Diffusion-Forcing) on the standard maze offline benchmarks.
> - TDP surpasses AdaptDiffuser on both the standard benchmark (PnP) and the custom task (PnWP). It supports TDP’s test-time scalability and generalization capability.
> - TDP surpasses MCSS+SS on all test tasks. It shows that increasing the test-time computation budget does not always guarantee better performance.
>
> **1. TDP performance on the standard maze benchmark**
>    - pointmaze-{*medium*,*large*,*giant*}-navigate-v0: 100±0
>    - antmaze-{*medium*,*large*}-navigate-v0: 100±0
>    - antmaze-*giant*-navigate-v0: 98±6
>
> TDP consistently outperforms both MCTD and Diffusion-Forcing across all standard maze benchmarks (see Table 1 in [b]).
>
> **2. Additional baselines on our benchmarks**
>
> We added two baselines: AdaptDiffuser for the Kuka tasks and MCSS+SS for all test tasks. AdaptDiffuser is a fine-tuned diffusion planner on synthetic expert demonstrations [32], and MCSS+SS refers to selecting the best trajectory from a batch generated via stochastic sampling [c]. MCSS+SS requires 4 times more computation compared to MCSS.
> - Gold-picking
> |**Environment**|**MCSS+SS**|**TDP**|
> |:-|-|-|
> |Maze2D-*Medium*|17.4±3.2|**39.8** ± 4.2|
> |Maze2D-*Large*|21.2±3.5|**47.6** ± 4.1|
> |Multi2D-*Medium*|29.2±3.6|**74.7** ± 3.0|
> |Multi2D-*Large*|58.0±3.8|**70.0** ± 3.5|
> - Pick-and-Place (PnP) and Pick-and-Where-to-Place (PnWP)
> |**Task**|**AdaptDiffuser**|**MCSS+SS**|**TDP**|
> |:-|-|-|-|
> |PnP (*stack*)|60.54 ± 0.18|56.8 ± 0.16|**61.17** ± 0.24|
> |PnP (*place*)|36.17 ± 0.11|35.5±0.19|**36.94** ± 0.13|
> |PnWP|39.72 ± 0.08|36.24 ± 0.09|**66.81**± 0.17|
> - AntMaze multi-goal exploration
> |**Metric**|**MCSS+SS**|**TDP**|
> |:-|-|-|
> |# found goals &uparrow;|62.8±1.9|**66.1** ± 1.9|
> |sequence match &uparrow;|31.2±1.3|**33.8** ± 1.4|
> |# timesteps per goal &downarrow;|604.4|**558.4**|
>
> Our results consistently show TDP's superior performance across all compared benchmarks.
>
> **3. Justification for the custom benchmarks**
>
> Recent diffusion planner approaches focus on **supervised** planning, including sequential methods [a, b], hierarchical methods [31, 4], and fine-tuning approaches [32] for standard offline RL benchmarks.
>
> TDP enhances **zero-shot** planning capability on unseen tasks *shifted* from the trained distribution. Given a test-time defined task, TDP operates as a task-aware planning framework on top of the pretrained diffuser. For example, in the AntMaze environment, the pretrained planners themselves hardly generalize to test-time multi-goal reward tasks since they are trained on the standard offline benchmarks which are single-goal tasks. The gold-picking task is also not solvable with a pretrained diffuser alone, as training trajectories lack intermediate goal conditioning. These tasks can be solved by directly applying TDP in a plug-and-play manner.
>
> Our benchmarks evaluate the zero-shot planning capability where planner models must infer high-reward trajectories purely from the guide function without expert demonstrations. Existing supervised planners challenge to solve our benchmarks. For instance, sequential approaches (e.g., MCTD) cannot outperform TDP on the PnWP task since MCTD explores *single-step* with guide/no-guide action, while TDP explores *multi-step* ahead with diverse bi-level trajectory sampling. Single-step exploration suits standard offline benchmarks with a single optimal goal, but often leads to local optima in multi-goal settings.
>
> > **Heuristic state decomposition** ...
>
> We would like to clarify that the observation-control state decomposition is neither heuristic nor manually specified. It is a **fully automated**, task-aware procedure performed at planning time. Given a guide function for the test-time task, states are decomposed based on gradient signals: observation states are those receiving non-zero gradients, while control states receive none. This gradient-based criterion enables scalable and domain-agnostic decomposition. We apologize for not making this process sufficiently clear in the manuscript. We will revise the corresponding method section to address this oversight.
>
> > **Ad-hoc dual guidance scheme** ...
>
> We would like to clarify that TDP does not implement a dual guidance scheme. Instead, TDP formulates a single joint conditional distribution with an integrated guidance term. As shown in Equation (2) of the main paper, the overall guidance distribution is defined as $h=h\_{gg}\cdot h\_{pg}$ where $h\_{gg}(\cdot)$ denotes the gradient guidance component, and $h\_{pg}(\cdot)$ denotes the particle guidance component. Since both components jointly condition the same pretrained reverse denoising process, the perturbed reverse denoising process is approximated as $\tilde{p}(\boldsymbol{\tau}\_{i-1}|\boldsymbol{\tau}\_i)\approx \mathcal{N}(\boldsymbol{\tau}\_{i-1}; \mu+\Sigma g\_{tdp}, \Sigma)$ where $g\_{tdp}=\nabla_\tau \log\left(h_{gg}(\boldsymbol{\tau}_i)\cdot h\_{pg}(\boldsymbol{\tau}\_i)\right)=g\_{gg}+g\_{pg}$. Intuitively, Equation (3) can be integrated to $\boldsymbol{\mu}^{i-1} \leftarrow \boldsymbol{\mu}^{i-1} + \Sigma^i g\_{tdp}$ where $g\_{tdp}=g\_{gg}+g\_{pg}= \alpha\_p\nabla \Phi(\boldsymbol{\mu}^{i-1}\_{\text{control}})+\alpha\_g\nabla \mathcal{J}(\boldsymbol{\mu}^{i-1}\_{\text{obs}})$. We have a single integrated guidance term, which is the sum of gradient guidance part ($g\_{gg}$) and particle guidance part ($g\_{pg}$).
>
> > **Insufficient methodological clarity** ...
>
> We appreciate this feedback and recognize that the overall pipeline could be clearer. Below, we provide a step-by-step description of how TDP operates. While individual components are described in lines 162-200, readers may benefit from seeing how these components work together.
>
> **Step 1. Parent Branching**: In the first phase of TDP, $N$ parent trajectories are generated via Equation (3). $N$ denotes the number of trajectories sampled in a batch.
> **Step 2. Sub-Tree Expansion**: In the second phase of TDP, $N$ child trajectories are generated via Equation (4). Each child trajectory is generated from a parent trajectory, and the branching site is randomly chosen among intermediate states in the parent trajectory.
> **Step 3. Leaf Evaluation**: Now we construct a trajectory tree consisting of parent and child trajectories. The root node of the tree corresponds to the initial state, and the tree has $2N$ leaf nodes. Each leaf node represents a complete trajectory and is associated with a guide score of the path from the root. The path to the leaf with the highest score is selected as the final solution trajectory.
> **Step 4. Action Execution**: TDP supports both open-loop and closed-loop planning. In open-loop planning, the agent executes the entire solution trajectory as planned. In closed-loop planning, the agent executes only the first action of the planned trajectory, then replans by repeating Steps 1-3 at each timestep.
>
> > **Task-specific reward engineering** ...
>
> We clarify that the notion of **“tasks”** mentioned by the reviewer differs from our problem setting. The tasks referenced by the reviewer correspond to the training tasks that a pretrained diffusion planner is expected to handle. While pretrained diffusion planners model underlying system dynamics, TDP samples high-reward solution trajectories (i.e., those with high guide scores for the test task) conditioned on the learned dynamics in a zero-shot manner. TDP equips the pretrained diffuser with the ability to reason over higher-level objectives, such as our benchmarks.
>
> Although the reviewer is correct that reward shape affects the performance of TDP, this is inherent to most zero-shot planning methods that rely on reward-guided adaptation at test time. Similar to prior works on zero-shot task transfer [d, e], TDP assumes access to a pretrained model (i.e., Diffuser) and dense reward signals to effectively adapt to unseen tasks.
>
> While TDP requires task-specific guide functions, it provides a unified framework that works across diverse domains without requiring task-specific model retraining or expert demonstrations.
>
> > ### Response to Q1 (Definition of trajectory quality)
>
> In our framework, trajectory quality is defined as a composite metric: the product of the guide reward J and a binary dynamic feasibility indicator. That is, a trajectory must not only align well with the test-time guide function but also be dynamically feasible (i.e., executable without violating environmental constraints).
>
> > ### Response to Q2 (Learned vs fixed PG)
>
> Learned potentials provide marginal improvements in both sample diversity and performance. The results below show TDP with learned vs fixed potentials evaluated on the Kuka benchmarks.
> - **Learned PG vs. Fixed PG**
>   - pairwise trajectory distance: Learned PG (17.54±0.04) vs. Fixed PG (17.38±0.05)
>   - PnP (*stack*) / PnP (*place*) / PnWP: Learned PG (61.28±0.22 / 37.04±0.12 / 67.23±0.15) vs. Fixed PG (61.17±0.24 / 36.94±0.13 / 66.81±0.17)
>
> [a] Boyuan Chen et al. Diffusion Forcing: Next-token Prediction Meets Full-Sequence Diffusion. 2024.
> [b] Jaesik Yoon et al. Monte Carlo Tree Diffusion for System 2 Planning. 2025.
> [c] Yanwei Wang et al. Inference-Time Policy Steering through Human Interactions. 2025.
> [d] Wenlong Huang et al. Language Models as Zero-Shot Planners: Extracting Actionable Knowledge for Embodied Agents. 2022.
> [e] Junhyuk Oh et al. Zero-Shot Task Generalization with Multi-Task Deep Reinforcement Learning. 2017.

---

> > ### Comment · Area_Chair_oiz2 · 2025-08-06
> > **Post-rebuttal**
> >
> > Dear reviewer gnZK,
> >
> > It would be appreciated if you can engage in the discussion and provide your input after the rebuttal from the authors or at least acknowledge that you have read the rebuttal, particularly given that this paper might be borderline. Pease also read the reviews+rebuttal from other reviewers and mention whether and why you keep your score.
> >
> > Best regards,
> > AC

---

> > ### Comment · Reviewer_gnZK · 2025-08-06
> >
> > Thank you for taking the time to provide such a thorough rebuttal. I especially appreciate the additional clarifications on methodological transparency and the detailed experimental results. Many of my initial concerns have been addressed, and I appreciate the authors’ effort in this regard.
> >
> > That said, some ambiguities remain:
> > In the rebuttal, you state that “It is a fully automated, task-aware procedure performed at planning time.” Could you indicate exactly where this is described in the manuscript? In particular, lines 162–172 provide only a limited explanation of how the two state sets are automatically separated. Without a clear revision to describe this procedure in detail, it is difficult to identify the precise starting point of the method. At a minimum, a concrete example from one task that illustrates the step-by-step operation of this automated system would be highly helpful.
> >
> > As a minor request, I would also appreciate it if the notation in the rebuttal were unified with that in the paper. The inconsistency is a bit confusing.
> >
> > Concerns that have been resolved:
> > 1. Narrow experimental coverage
> > The additional results show TDP outperforming MCTD and Diffusion Forcing, and producing superior results on PnP and PnWP. In particular, the inclusion of OGBench experiments, as I requested, and the fact that TDP outperforms MCTD is impressive. The results against AdaptDiffuser and MCSS+SS are also convincing. I consider this concern resolved.
> >
> > 2. Insufficient methodological clarity
> > This issue has also been resolved. If the paper is accepted, I suggest moving the step-by-step description to the beginning of the method section so readers can more easily follow the pipeline.
> >
> > 3. Ad-hoc dual guidance scheme
> > I agree that treating the two components as a product to form a single guidance distribution is a reasonable interpretation. However, I still understand that $\mu_\text{control}$ and $\mu_\text{obs}$ are disjoint sets of elements, with separate guidance applied to each. Unless the fully automated process for partitioning these elements is clearly provided, I cannot fully resolve this concern. This may stem from my incomplete understanding of the proposed state decomposition, and I would appreciate a more detailed explanation or clarification in the final version.
> >
> > 4. Task-specific reward engineering
> > The authors’ explanation on this point was highly persuasive.
> >
> > Since three of my four concerns have been fully resolved, I will raise my score to 3. I lean towards borderline reject, but I will not strongly oppose acceptance.

---

> > > ### Author Response · Authors · 2025-08-07
> > >
> > > > I would also appreciate it if the notation in the rebuttal were unified with that in the paper. The inconsistency is a bit confusing.
> > >
> > > We sincerely apologize for the inconsistent notation between the rebuttal and the paper. We will unify the notation according to the following criteria:
> > >
> > > 1. $\alpha$ and $g$: To resolve inconsistencies between Equation (2) and our rebuttal regarding guidance scale and guidance components, we will explicitly separate these notations in all equations to indicate the guidance scale for each guidance scheme. Therefore, the equation in our rebuttal will use $\alpha\_{tdp}$ to denote the guidance scale for TDP.
> > >
> > > 2. $\Sigma$ and $\Sigma^i$: To address inconsistent covariance notation between Equations (2) and (3), we will update Equation (2) to follow the same notation ($\Sigma \rightarrow \Sigma^i$).
> > >
> > > 3. $\boldsymbol{\mu}^{i}$ and $\boldsymbol{\mu}^{i-1}$: To avoid confusion in the notation when applying guidance to the diffusion-predicted mean $\boldsymbol{\mu}$ for each denoising step, we will consistently denote $\boldsymbol{\mu}^{i-1}$ as the one-step guided predicted mean from $\boldsymbol{\mu}^{i}$ for all guidance schemes.
> > >
> > >
> > > **Updated equations:**
> > > - Paper equations:
> > >   - Equation (2): $\tilde{p}(\boldsymbol{\tau}\_{i-1}|\boldsymbol{\tau}\_i) \propto p\_\theta(\boldsymbol{\tau}\_{i-1}|\boldsymbol{\tau}\_i)h(\boldsymbol{\tau}\_i) \approx \mathcal{N}(\boldsymbol{\tau}\_{i-1}; \mu^i+\alpha\Sigma^i g, \Sigma^i)$
> > >
> > >   - Equation (3): $\boldsymbol{\mu}^{i-1}\_{\text{control}} \leftarrow \boldsymbol{\mu}^{i}\_{\text{control}} + \alpha_p \Sigma^i \nabla \Phi(\boldsymbol{\mu}^{i}\_{\text{control}})$, $\boldsymbol{\mu}^{i-1}\_{\text{obs}} \leftarrow \boldsymbol{\mu}^{i}\_{\text{obs}} + \alpha_g \Sigma^i \nabla \mathcal{J}(\boldsymbol{\mu}^{i}\_{\text{obs}})$
> > >
> > >   - line 184: $\boldsymbol{\mu}^{i-1}=[\boldsymbol{\mu}^{i-1}_{\text{control}}, \boldsymbol{\mu}^{i-1}\_{\text{obs}}]$
> > >
> > > - Rebuttal equations:
> > >   - $\tilde{p}(\boldsymbol{\tau}\_{i-1}|\boldsymbol{\tau}\_i)\approx \mathcal{N}(\boldsymbol{\tau}\_{i-1}; \mu^i+\alpha\_{tdp}\Sigma^i g\_{tdp}, \Sigma^i)$
> > >   - $\boldsymbol{\mu}^{i-1} \leftarrow \boldsymbol{\mu}^{i} + \alpha\_{tdp}\Sigma^i g\_{tdp}$
> > >
> > >
> > > We will also update the notation in Algorithm 2 according to the suggested criteria to ensure consistency throughout the manuscript. We appreciate your careful review and attention to detail, which helps us improve the overall clarity and quality of our work.

---

> ### Author Response · Authors · 2025-08-07
>
> We greatly appreciate your constructive feedback and your decision to increase the score. We are pleased that most of the concerns have been resolved during the rebuttal process. We would like to address the remaining concern with the following clarifications.
>
> > In the rebuttal, you state that “It is a fully automated, task-aware procedure performed at planning time.” Could you indicate exactly where this is described in the manuscript? … Without a clear revision to describe this procedure in detail, it is difficult to identify the precise starting point of the method.
>
> We apologize for not clearly indicating the autonomous nature of the state decomposition in the manuscript. To clarify this process, we provide Algorithm 4 **State Decomposition** below.
>
> 1: &nbsp;&nbsp;**Input:** task guide $\mathcal{J}$, trajectory $\boldsymbol{\tau}$
>
> 2: &nbsp;&nbsp;$W \coloneqq $ number of features in state vector of $\boldsymbol{\tau}$
>
> 3: &nbsp;&nbsp;$(\boldsymbol{s}_1, \boldsymbol{s}_2, … \boldsymbol{s}_W) \coloneqq \boldsymbol{\tau}$
>
> 4: &nbsp;&nbsp;$l_{\text{control}} \leftarrow [], l_{\text{obs}} \leftarrow []$
>
> 5: &nbsp;&nbsp;**for** $i = 1$ to $W$ **do**
>
> 6: &nbsp;&nbsp;&nbsp;&nbsp;&nbsp;&nbsp; **if** $\frac{\partial \mathcal{J}}{\partial \boldsymbol{s}_i} == 0$ **then**
>
> 7: &nbsp;&nbsp;&nbsp;&nbsp;&nbsp;&nbsp;&nbsp;&nbsp;&nbsp;&nbsp; Append $\boldsymbol{s}\_i$ to $l_{\text{control}}$
>
> 8: &nbsp;&nbsp;&nbsp;&nbsp;&nbsp;&nbsp; **else**
>
> 9: &nbsp;&nbsp;&nbsp;&nbsp;&nbsp;&nbsp;&nbsp;&nbsp;&nbsp;&nbsp; Append $\boldsymbol{s}\_i$ to $l_{\text{obs}}$
>
> 10: &nbsp;&nbsp;&nbsp;&nbsp; **end if**
>
> 11: **end for**
>
> 12: **return** $[l_{\text{control}}, l_{\text{obs}}]$
>
> Accordingly, line 4 in Algorithm 2 should be replaced by $[\mu_{\text{control}}, \mu_{\text{obs}}] \leftarrow \text{State Decomposition}(\mathcal{J}, \mu_\theta(\boldsymbol{\tau}_{\text{parent}}^i))$.
>
> > At a minimum, a concrete example from one task that illustrates the step-by-step operation of this automated system would be highly helpful.
>
> For instance, the KUKA robot arm environment provides state vectors containing multiple features such as robot joint angles and block positions. Since TDP operates as a zero-shot planner, it does not have prior knowledge of the state category for each feature of the state vector. Given a test-time block stacking task with a distance-based guide function, TDP autonomously categorizes each feature value in the state vector using Algorithm 4 (State Decomposition). It evaluates whether the gradient of the guide function with respect to the *i*th feature (i.e., $\frac{\partial \mathcal{J}}{\partial \boldsymbol{s}_i}$)  is zero or non-zero. If non-zero, the *i*th feature is classified as an *observation* state; if zero, it is classified as a *control* state. Consequently, features related to robot physics are detected as *control* states, while block position (xy) features are detected as *observation* states, because the block-stacking guide function is only affected by the position of the blocks.
>
> We acknowledge that the current manuscript presents the *observation*-*control* state decomposition as if it were pre-determined based on each feature's fundamental properties, which may have caused confusion. We will carefully revise the text to reflect the autonomous nature of the state decomposition process as described in our rebuttal.
>
> > However, I still understand that $\mu_\text{control}$ and $\mu_\text{obs}$ are disjoint sets of elements, with separate guidance applied to each. Unless the fully automated process for partitioning these elements is clearly provided, I cannot fully resolve this concern.
>
> TDP's task-aware guidance scheme consists of gradient guidance (for task-relevant features) and particle guidance (for task-independent features). As explained above regarding Algorithm 4, *control* states, which represent task-independent features, are autonomously detected based on the input task ($\mathcal{J}$). TDP applies particle guidance to these *control* states, which are not influenced by gradient guidance. While gradient guidance is applied to all features, only *observation* states, which represent task-relevant features, are affected by this term. Therefore, TDP's integrated guidance term serves two complementary functions: promoting diversity in *control* states while simultaneously optimizing *observation* states based on the task guide function.

---

> ### Comment · Reviewer_gnZK · 2025-08-08
>
> Through the authors' thorough rebuttal and discussion, all of my concerns have been addressed. I hope that the strengthened experimental results and clearer explanations will be reflected in the next revision, and I am raising my score to a 4.

---

> > ### Author Response · Authors · 2025-08-08
> >
> > Thank you very much for carefully reconsidering your evaluation after our rebuttal and discussion. We truly appreciate your openness to updating your assessment in a positive direction. Your feedback has been invaluable in helping us improve both the clarity of our explanations and the coverage of our experimental results, which we will ensure are reflected in the next revision.

---

### Official Review · Reviewer_k36W · 2025-07-03

**Clarity:** 3
**Significance:** 3
**Originality:** 3
**Rating:** 4
**Confidence:** 3

**Summary:**

In this work, the authors propose a Tree-guided Diffusion Planner (TDP) that addresses challenges posed by non-convex objectives, non-differentiable constraints, and multi-reward scenarios. The method first decomposes the state into observation and control components. It then combines particle guidance with gradient-based guidance and employs a bi-level sampling scheme, where multiple child trajectories are sampled at each timestep. This design increases the likelihood of reaching global optima, avoiding the local traps that commonly hinder traditional gradient-based methods. Experimental results demonstrate that TDP outperforms state-of-the-art diffusion planning approaches.

**Questions:**

1. The examples of non-convex objectives or non-differentiable constraints in the supplementary material appear solvable via subgoals or intermediate rewards using hierarchical diffusion. Does TDP offer specific advantages over such hierarchical approaches? Or are there cases where hierarchical methods fail but TDP works?

2. The paper does not provide sufficient detail on how particle guidance is computed. How is the particle guidance calculated? Is it applied only to the control states, or to the original state?

3. How are observation and control states separated? Are they identified through automated method (e.g., gradient-based analysis) or specified manually?

**Ethical Concerns:**

["NO or VERY MINOR ethics concerns only"]

**Final Justification:**

I thank the authors for the detailed responses during the discussion phase and for outlining clear plans to improve the paper. I remain positive about its acceptance. However, since I cannot evaluate the extent of the planned revisions, I am unable to increase my score and will maintain my current evaluation.

**Limitations:**

The limitations are addressed in the conclusion section.

**Paper Formatting Concerns:**

There are no major formatting issues in the paper.

**Quality:**

3

**Strengths And Weaknesses:**

Strengths:

1. The authors propose a novel tree-guided method that increases the probability of reaching the global optimum compared to traditional gradient-based approaches.

2. Theoretical analysis is provided to support the main proposition, offering insight into the method’s effectiveness.

3. The experimental results are comprehensive, covering multiple tasks and clearly demonstrating the method’s effectiveness and reliability.

4. Detailed implementation information is provided in the appendix, which enhances reproducibility.

Weaknesses:

1. I strongly recommend the authors to split the method section into separate subsections to improve clarity and better highlight the contributions of each component.

2. The sub-tree expansion procedure seems very similar to conventional MCSS method,  which may reduce the novelty of the proposed approach.

---

> ### Author Rebuttal · Authors · 2025-07-29
>
> We greatly appreciate Reviewer **k36W** for the thoughtful comments. Below, we address each of the raised concerns.
>
> > I strongly recommend the authors to split the method section into separate subsections to improve clarity and better highlight the contributions of each component.
>
> As the reviewer's suggestion, we will revise the method section (Sec. 3) into three subsections (State Decomposition, Parent Branching, Sub-Tree Expansion) to clarify the contributions of each component of TDP. Additionally, we provide a detailed summary of each component’s contribution.
> **Sec 3.1 State Decomposition**: The *observation*-*control* state decomposition is a **task-aware** and fully **automated** procedure performed at planning time. Given a guide function for the test-time task, states are decomposed based on gradient signals: *observation* states are those receiving non-zero gradients, while control states receive none. Since this decomposition relies solely on the task definition, it requires no prior knowledge or manual specification of the state components.
> **Sec 3.2 Parent Branching**: In this first procedure of our bi-level sampling, control states are applied to particle guidance in order to **explore diverse *control* trajectories**. For example, when we want to move a block to a target position, there are many possible *control* trajectories to succeed in this task. While the traditional gradient-based approaches meet challenges (i.e., in-distribution preference, insufficient exploration) to generate diverse trajectories, TDP enables enhanced exploration via this procedure. We refer to these trajectories as parent trajectories.
> **Sec 3.3 Sub-Tree Expansion**: In this second procedure of our bi-level sampling, parent trajectories are partially refined via fast-denoising. Each child trajectory is generated from a parent trajectory, and the branching site is randomly chosen among intermediate states. We refer to these trajectories as child trajectories. Sub-tree expansion offers two key advantages:
> - Enhance **dynamic feasibility** of parent trajectories: Diverse parent trajectories benefit exploration, but perturbing the *control* states may lead to dynamically infeasible plans. During sub-tree expansion, perturbed *control* states are refined by a pretrained diffusion denoising process.
> - Efficient **Local search**: Sub-Tree expansion refines *observation* states of parent trajectories with gradient guidance signal. Parent trajectories serve as initial points to guide child trajectories. Since parent trajectories are intended to cover a broad region of search space, local search conditioned on the parent trajectories is an efficient way to find better local optima.
>
> The contribution of each component of TDP is empirically validated across multiple environments, where TDP consistently outperforms both ablated variants (TDP (w/o child) and TDP (w/o PG)) on all test tasks.
>
> > The sub-tree expansion procedure seems very similar to conventional MCSS method, which may reduce the novelty of the proposed approach.
>
> We would like to clarify that TDP's sub-tree expansion procedure is distinct from conventional MCSS methods in the following two key aspects.
> - Novel **initialization scheme** based on theoretical analysis: TDP leverages diverse parent trajectories as initialization points for generating child trajectories via a diffusion process. This enables sub-tree expansion to perform local search from multiple, sparsely distributed points across the state space. In contrast, MCSS typically uses zero-centered noisy initializations, which are more likely to converge to local optima, as highlighted in Proposition 1-a.
> - **Temporal exploration** via random branching: Sub-tree expansion selects a random branching site for each trajectory. As a zero-shot planner, TDP receives no expert demonstrations and must instead infer optimal behavior through the guide function alone. By uniformly sampling the branching site (timestep) from the range $[0, T_{\text{pred}}]$ where $T_{\text{pred}}$ is the planning horizon, TDP can effectively discover temporally distant optimal goals. In contrast, MCSS lacks this mechanism and may fail to reach such goals—particularly those requiring progress beyond a certain timestep threshold—due to its concurrent sampling scheme [23].
>
> Furthermore, we empirically evaluate the scalability of MCSS under increased computational budgets on our tested benchmarks. Stochastic Sampling (SS) [b] incrementally enhances trajectory quality by using diffusion steps that are 4 times longer than the original. **MCSS+SS** denotes selecting the best trajectory from a batch generated via this stochastic sampling process. As shown in the results below, TDP consistently outperforms MCSS+SS across all test tasks, reaffirming the limited test-time scalability and generalization capacity of MCSS in comparison to TDP.
>
> - Maze2D gold-picking
> |**Environment**|**MCSS+SS**|**TDP**|
> |:-|-|-|
> |Maze2D-*Medium*|17.4±3.2|**39.8** ± 4.2|
> |Maze2D-*Large*|21.2±3.5|**47.6** ± 4.1|
> |Multi2D-*Medium*|29.2±3.6|**74.7** ± 3.0|
> |Multi2D-*Large*|58.0±3.8|**70.0** ± 3.5|
> - Pick-and-Place (PnP) and Pick-and-Where-to-Place (PnWP)
> |**Task**|**MCSS+SS**|**TDP**|
> |:-|-|-|
> |PnP (*stack*)|56.8 ± 0.16|**61.17** ± 0.24|
> |PnP (*place*)|35.5±0.19|**36.94** ± 0.13|
> |PnWP|36.24 ± 0.09|**66.81**± 0.17|
> - AntMaze Multi-goal Exploration
> |**Metric**|**MCSS+SS**|**TDP**|
> |:-|-|-|
> |# found goals &uparrow;|62.8±1.9|**66.1** ± 1.9|
> |sequence match &uparrow;|31.2±1.3|**33.8** ± 1.4|
> |# timesteps per goal &downarrow;|604.4|**558.4**|
>
> > ### Response to Q1 (TDP vs hierarchical methods)
>
> Our evaluated benchmarks are not solvable using standard hierarchical approaches. Below, we clarify the fundamental differences between TDP and hierarchical diffusion planners from both the modeling and benchmark perspectives.
> **1. Model-aspect Distinction**
> While hierarchical diffusion planners boost **supervised** planning capability with improved compositional behaviors through the use of subgoals and intermediate rewards [a, 31, 4], TDP boosts **zero-shot** planning capability on top of pre-trained diffusion planners. Hierarchical methods excel in generalizing over **labeled** data distributions seen during training. In contrast, TDP is designed to solve **unseen** planning tasks where a guide function is provided only at test time. As a bi-level sampling framework, TDP explores the search space in a compositional manner, enabling efficient discovery of high-reward trajectories even in previously unseen task configurations.
>
> **2. Benchmark-aspect Distinction**
> The benchmark suites we evaluate consist entirely of **test-time labeled** tasks that were not seen during training, posing a challenge for hierarchical planners that rely on training-time supervision. For example, while prior work [4] demonstrates that hierarchical diffusion planners can effectively generate *shortest-path* trajectories when both initial and goal states are specified, our gold-picking benchmark presents a fundamentally different challenge: the goal is *hidden* and often not aligned with the shortest path.
>
> Furthermore, the gold-picking task does not provide explicit information about intermediate goals (e.g., the location of the gold), nor is it guaranteed that such a goal even exists. Similarly, the PnWP task offers no precise positional information for either of the two optimal targets (i.e. local and global). As a result, the planner must infer high-reward trajectories purely based on the guide function. While some example trajectories in the supplementary material may appear compositional, these emerge from TDP’s bi-level search rather than explicit supervision. Hierarchical approaches are unlikely to succeed on these benchmarks without incorporating task-specific heuristics or annotations.
>
> > ### Response to Q2 (PG implementation details)
>
> The particle guidance term is calculated **only** with the ***control* states**. From line 173 to line 189 in the paper, we describe how particle guidance is computed. Specifically, this term is equal to the gradient of a radial basis function (RBF), denoted as $\nabla\Phi$, which can be directly computed with all pairwise distances among *control* trajectories within a batch. For clarity, this process is also illustrated in line 6 of Algorithm 2.
>
> > ### Response to Q3 (Observation and control decomposition)
>
> The observation and control states are **autonomously decomposed** at test time through binary gradient checking with respect to the inferred task guide function. This process requires no manual intervention or prior domain knowledge. We have previously addressed this in our initial response.
>
> [a] Jinning Li et al. Hierarchical Planning Through Goal-Conditioned Offline Reinforcement Learning. 2022.
> [b] Yanwei Wang et al. Inference-Time Policy Steering through Human Interactions. 2025.

---

> > ### Comment · Reviewer_k36W · 2025-08-05
> >
> > I thank the authors for their responses and detailed plans for revising the paper. I will maintain my current evaluation and continue following the discussions from the other reviewers.

---

> > > ### Author Response · Authors · 2025-08-05
> > >
> > > We sincerely thank you for taking the time to review our responses and for acknowledging our revision plans. We look forward to incorporating all reviewers' feedback in the final manuscript. Thank you for your continued consideration.

---

> > > ### Author Response · Authors · 2025-08-08
> > >
> > > Dear Reviewer **k36W**,
> > >
> > > We sincerely thank you for your time and engagement throughout the review process.
> > >
> > > We understand your earlier concern regarding novelty, and would like to highlight that our approach incorporates several distinctive elements — including a novel **initialization scheme** grounded in theoretical analysis and **temporal exploration** via random branching — which make TDP's sub-tree expansion fundamentally different from MCSS.
> > >
> > > With the concerns from the other reviewers now resolved, we hope these aspects will also be taken into consideration when you make your final evaluation.
> > >
> > > Best regards,
> > > The Authors of Submission #26284

---

### Note · Authors · 2025-08-12

We thank the reviewers and AC for their constructive and active engagement throughout the process, which greatly strengthened this work. Following the rebuttal and discussions, all major concerns have been resolved:

- **Methodological clarification**: Clarified that observation–control state decomposition is a fully autonomous procedure (Algorithm 4) rather than a heuristic; explained the integrated use of gradient and particle guidance in parent branching for improved exploration; revised the Method section by splitting it into three clear subsections for enhanced clarity and presentation.
- **Recent related work**: Expanded discussion of sequential approaches, hierarchical approaches, and Diffusion-MPC.
- **Experimental coverage**: Added baselines (AdaptDiffuser, MCSS+SS); included TDP results on standard offline maze benchmarks; provided extensive ablations (gradient/particle guidance strength $\alpha\_g / \alpha\_p$, fast denoising steps $N_f$, fixed vs. learned PG).
- **Performance justification**: Provided analysis clarifying why TDP outperforms MCTD, even in single-task settings; explained why our tested benchmarks are not solvable by standard hierarchical approaches.
- **Tested benchmark justification**: Explained that TDP is a zero-shot planner, while many recent approaches are supervised planners requiring task-specific training, motivating our benchmark choice.
- **Reproducibility**: Confirmed full code release upon acceptance

During the discussion period, Reviewer **gnZK** raised their score from 2 $\rightarrow$ 4, Reviewer **GPcL** from 3 $\rightarrow$ 4, and Reviewer **2mBS** from 3 $\rightarrow$ 4 after our clarifications. Reviewer **k36W** maintained a 4. All reviewers now hold positive scores ($\geq$4), and none oppose acceptance.

We believe the strengthened explanations, broader experiments, and richer contextualization significantly improve the work. We remain committed to releasing the complete code and benchmarks upon acceptance, as well as revising the manuscript in accordance with the detailed plan outlined in our rebuttal, to ensure full reproducibility and clarity.

---

### Decision · Program_Chairs · 2025-09-17

**Decision:**

Accept (poster)

**Comment:**

The paper is judged good from the theoretical and experimental perspectives by all reviewers. The paper has (or at least had before the rebuttal) some weaknesses and concerns that have been addressed during the rebuttal phase:
- The clarity of the methodology was discussed (Reviewer gnZK and to some degree Reviewer k36W),
- The relatively limited novelty was mentioned either explicitly or through comments about some other related work.
- There were mixed opinions on the experimental scope. Reviewer gnZK considered them as narrow while Reviewer GPcL considered them as extensive even though not on large tasks.

Overall, the rebuttal from the authors convinced all reviewers to either stay or move to a score of 4 (weak accept). Therefore the recommendation is to accept the paper.